# The Pancreatic β-Cell: The Perfect Redox System

**DOI:** 10.3390/antiox10020197

**Published:** 2021-01-29

**Authors:** Petr Ježek, Blanka Holendová, Martin Jabůrek, Jan Tauber, Andrea Dlasková, Lydie Plecitá-Hlavatá

**Affiliations:** Department of Mitochondrial Physiology, No.75, Institute of Physiology of the Czech Academy of Sciences, 14220 Prague, Czech Republic; blanka.holendova@fgu.cas.cz (B.H.); martin.jaburek@fgu.cas.cz (M.J.); jan.tauber@fgu.cas.cz (J.T.); andrea.dlaskova@fgu.cas.cz (A.D.); lydie.plecita@fgu.cas.cz (L.P.-H.)

**Keywords:** pancreatic β-cells, insulin secretion, redox signaling, NADPH oxidase 4, branched-chain ketoacid oxidation, fatty-acid-stimulated insulin secretion, ATP-sensitive K^+^ channel, TRPM channels, GLP-1, GPR40

## Abstract

Pancreatic β-cell insulin secretion, which responds to various secretagogues and hormonal regulations, is reviewed here, emphasizing the fundamental redox signaling by NADPH oxidase 4- (NOX4-) mediated H_2_O_2_ production for glucose-stimulated insulin secretion (GSIS). There is a logical summation that integrates both metabolic plus redox homeostasis because the ATP-sensitive K^+^ channel (K_ATP_) can only be closed when both ATP and H_2_O_2_ are elevated. Otherwise ATP would block K_ATP_, while H_2_O_2_ would activate any of the redox-sensitive nonspecific calcium channels (NSCCs), such as TRPM2. Notably, a 100%-closed K_ATP_ ensemble is insufficient to reach the −50 mV threshold plasma membrane depolarization required for the activation of voltage-dependent Ca^2+^ channels. Open synergic NSCCs or Cl^−^ channels have to act simultaneously to reach this threshold. The resulting intermittent cytosolic Ca^2+^-increases lead to the pulsatile exocytosis of insulin granule vesicles (IGVs). The incretin (e.g., GLP-1) amplification of GSIS stems from receptor signaling leading to activating the phosphorylation of TRPM channels and effects on other channels to intensify integral Ca^2+^-influx (fortified by endoplasmic reticulum Ca^2+^). ATP plus H_2_O_2_ are also required for branched-chain ketoacids (BCKAs); and partly for fatty acids (FAs) to secrete insulin, while BCKA or FA β-oxidation provide redox signaling from mitochondria, which proceeds by H_2_O_2_ diffusion or hypothetical SH relay via peroxiredoxin “redox kiss” to target proteins.

## 1. Introduction

### 1.1. Emerging Concept of Redox Signaling

The emerging concept of redox signaling resulted from ceaseless efforts, and decades of studies of the redox homeostasis of cells and cell organelles. Among organelles, the mitochondrion combines the key roles of the metabolic and redox hub. Prototypical redox signaling is involved in mechanisms of hypoxia-inducible factor (HIF) system-initiated transcriptome reprogramming. Both cytosolic and mitochondrial redox signaling were implicated, e.g., in HIF-1α stabilization, either via the oxidation of iron II to iron III of the reaction center of prolyl hydroxylase domain enzymes (PHD/EglN) [1,2] and/or their sensitive cysteines [3].

Recently, we revealed another physiological redox signaling of enormous biomedical significance: NADPH oxidase 4- (NOX4-) initiated redox signaling (H_2_O_2_), which fundamentally determines the 1st phase of glucose-stimulated insulin secretion (GSIS) in pancreatic β-cells, hypothetically by a cooperative closing of the ATP-sensitive K^+^ channel (K_ATP_) together with ATP elevation (Figure 1A) [4]; or alternatively (Figure 1B), we may assume that the essentially synergic channels, such as transient receptor potential melastin (TRPM) isoform 2, i.e., TRPM2, are also activated by H_2_O_2_ as reported [5], when the 100% K_ATP_ ensemble is closed by the elevated ATP.

The inability of the closure of the entire K_ATP_ ensemble to reach the threshold depolarization of the plasma membrane for opening voltage-gated Ca^2+^ channels (Ca_V_) [6] is discussed in this review as the key element for understanding the mechanism of GSIS. Thus synergy is required with the other channels. Moreover, the principle of a “logical summation” of metabolic plus redox stimulation will be discussed and the experimental evidence supporting it. This principle seems to be universal for any mechanism of insulin secretion dependent on K_ATP_ and/or Ca_V_.

For example the alternative, in this case mitochondrial redox signaling, is essentially required for insulin secretion stimulated by metabolites of branched-chain amino acids (BCAAs), i.e., BC keto acids (BCKAs), including 2-ketoisocaproate (KIC; termed also 2-oxoisocaproate, OIC; leucine metabolite), 2-ketoisovalerate (KIV; valine metabolite) and 2-ketomethylvalerate (KMV; isoleucine metabolite). Evidence for this mechanism stems from the effects of mitochondrial matrix-targeted antioxidant SkQ1, which does not affect GSIS in INS-1E cells, but completely inhibits insulin secretion stimulated by BCKAs, e.g., by KIC [4]. The interpretation of these results is that the NOX4 source of H_2_O_2_ cannot be easily inhibited by SkQ1 located within the inner phospholipid leaflet of the inner mitochondrial membrane (IMM), whereas the redox signaling originating from the mitochondrion must be blocked.

Previously, also coenzyme A-esters (CoA-esters) of fatty acids (FAs), malonyl-CoA and long-chain acyl-CoA were suggested to be coupling factors for nutrient stimulus-insulin secretion coupling [7]. Current complex knowledge on BCKAs and FAs in relation to insulin secretion with an emphasis on redox signaling is also reviewed here.

It has been established that the content of glutathione is relatively low in pancreatic β-cells [8,9,10,11], in contrast to the content of thioredoxins and glutaredoxins [12,13], peroxiredoxins and other proteins capable of redox relay, hence suitable for conducting and spreading redox signals [14,15]. From this point of view, the pancreatic β-cell appears to be a perfect redox system and ideally suited for spreading the redox signal.

Without knowing all the details of this redox system and the sensing of glucose or other secretagogues, one cannot judge their impairment during the development of type 2 diabetes. However, discussion of the resulting pathology is outside the scope of this review article, and the origin of oxidative stress in diabetic pancreatic β-cells, which has been reviewed elsewhere [16]. Neither do we discuss the relations to and effects of redox signaling on gene expression in pancreatic β-cells. The repeatable redox signaling in pancreatic β-cells, such as during GSIS, was suggested to maintain the correct expression of the insulin gene (*Ins*) [17]. The repeatable redox signaling could hypothetically maintain the other transcription factors or proteins specific for the optimum fitness and identity of β-cells. This vast area of knowledge is beyond the scope of this review.

### 1.2. Traditional Classification of Insulin Secretion Mechanisms

The consensus of reports on GSIS has been that it involves two classes of mechanisms or pathways [18,19,20,21,22,23,24,25,26]. By definition, triggering mechanism(s) were exclusively ascribed to the canonical one or those involving the K_ATP_-channel-dependent Ca^2+^ influx into β-cells provided by Ca_V_ opening. In turn, the metabolic amplifying mechanisms were defined as those involving mechanisms for receptor signaling that activate insulin secretion, including incretin receptor and metabotropic receptor signaling. The latter typically involves other secretagogues, such as FAs. Certain inconsistencies in such a classification are apparent. For example, upon the stimulation of insulin secretion by FAs at low, non-stimulating glucose, there are “mixed” mechanisms or pathways, which are both K_ATP_-channel dependent and receptor dependent. The latter is also dependent (but maybe only partially) on the Ca_V_-mediated Ca^2+^ influx into β-cells. It is questionable whether there are any mechanisms that do not involve the primary Ca_V_-mediated Ca^2+^ influx. Speculatively, such a net amplifying mechanism should be exclusively for the docking, fusion and exocytosis of insulin granule vesicles (IGVs) independent of Ca_V_-triggering. Hence, the responses of the exocytotic machinery should be given by the elevating cytosolic Ca^2+^ concentration [Ca^2+^]_c_, which is not triggered by the primary Ca_V_-mediated Ca^2+^ influx (as for the Ca^2+^ efflux from ER), but originates from signaling by protein kinase A (PKA; see Section 2.5.2), by a pathway of enhanced signaling via exchange proteins directly activated by cAMP 2 (EPAC2; see Section 2.5.3) and or G-protein-coupled receptors (GPR; see Section 2.5.5 and Section 6). Nevertheless, typically information signaling induces mixed responses involving both Ca_V_-triggering plus the facilitation of IGV exocytosis. Even if we do not know whether these two mechanism categories can exist separately, we use the term “Ca_V_-dependent“ or “Ca_V_- plus receptor-dependent“ insulin secretion, while the latter can be subclassified, whether it includes the Ca^2+^-efflux from the ER or not (Figure 2). The ER-Ca^2+^-independent mechanisms thus originate from the direct signaling of PKA, EPAC2 and GPR, exclusively affecting targets among the IGV-exocytotic machinery.

Note also that the two phases of insulin secretion occur in experiments with mice or their PIs, even for FAs [27]. As such, each phase could be based on its own distinct spectrum of mechanisms, being amplified “separately” and by a slightly distinct mechanism. That is why in this review, we instead describe and discuss insulin secretion according to the primary secretagogue.

## 2. Revisited Mechanism of Glucose-Stimulated Insulin Secretion

### 2.1. Elevations of ATP Plus H_2_O_2_ as the Fundamental Condition for GSIS

#### 2.1.1. Discovery of NOX4-Mediated Redox Signaling in Pancreatic β-Cells

Reactive oxygen species (ROS) have been implicated in insulin secretion. For example, mitochondrial ROS have been suggested to modulate insulin secretion [28], when resulting from mono-oleoyl-glycerol addition [29]. In Section 6 below, this is explained by the direct activation of GPR119 and possible fatty-acid- (FA-) stimulated insulin secretion (FASIS), while the observed ROS could originate from FA β-oxidation. Inhibition of the pentose phosphate pathway (PPP), which diminishes insulin secretion, was also reported to shift redox homeostasis [30]. Instead an unspecified link of GSIS with the externally added H_2_O_2_ was observed, and the effects of antioxidants at decreased glutathione by diethylmaleate in INS-1(823/13) cells [31]. The inhibition of an unidentified isoform of NADPH oxidase was also reported to attenuate GSIS when β-cells were treated with an antisense p47PHOX oligonucleotide [32] or with a nonspecific inhibitor of NADPH oxidase (DPI), which also inhibits complex I of the mitochondrial respiratory chain (RC) [33,34]. Another report employed an inhibitor of the two specific isoforms of NADPH oxidases (NOXs) [29].

Nevertheless, it was not known which NOX isoform actually participates in GSIS, or if any is involved at all. Thus, the traditional consensus on a “textbook” GSIS mechanism was maintained, which considers the ATP elevation (or elevation of the ATP/ADP ratio) to be the only requirement and a sufficient condition for triggering insulin secretion (e.g., by the K_ATP_ closing), with no need for any parallel redox signaling [18,19,20,21]. It had never been considered that the ATP increase by itself is insufficient for GSIS. In contrast, we demonstrated that the elevated oxidative phosphorylation (OXPHOS), leading to the increased ATP levels and ATP/ADP ratio at the peri-plasmamembrane space in the vicinity of K_ATP_, is insufficient to initiate GSIS [4]. Additionally, it had never been considered that any redox signaling might essentially participate in GSIS.

We have finally demonstrated that NADPH oxidase isoform 4 (NOX4) is fundamentally required for GSIS [4]. In model rat pancreatic β-cells (INS-1E cells) with silenced NOX4 or in full NOX4 knockout (NOX4KO) mice and in mice with NOX4 knockout specifically in pancreatic β-cells (NOX4βKO mice), the 1st phase of GSIS was largely blocked [4]. Unlike in INS-1E cells (where two phases of insulin secretion do not exist), in both NOX4 KO mice strains and in their isolated pancreatic islets (PIs), the 1st (fast) phase of GSIS was abolished with NOX4 ablation. In PIs, either the overexpression of NOX4 (achieved at least in the peripheral spheroid layer of islets) or additions of H_2_O_2_ rescued this 1st fast phase of GSIS. No effects were found in NOX2 KO mice, despite NOX2 having been previously implicated to play an antagonistic role in redox homeostasis [35]. In turn, similar effects were observed with doxycycline-inducible β-cell-specific NOX5 knockout mice [36].

Moreover, using the patch-clamp of INS-1E cells silenced for NOX4, no glucose-induced closure of a channel with all the characteristics of K_ATP_ was observed [4]. Simultaneously, in INS-1E cells transfected with scrambled siRNA, we confirmed the well-known closure of K_ATP_, as induced by glucose. These experiments supported the model in which K_ATP_ (not together with the essentially synergic channels, such as TRPM2) integrates metabolic and redox homeostasis, and the channel system acts as a logical summation for which both elevated ATP plus elevated H_2_O_2_ exclusively lead to the required plasma membrane depolarization in order to activate Ca_V_. Elevations of either ATP alone or H_2_O_2_ alone cannot establish such conditions (Figure 1 and Figure 2). The current most plausible hypotheses are that either both elevated ATP plus H_2_O_2_ act directly on K_ATP_ (Figure 1A), or alternatively the TRPM2-K_ATP_ synergy could be behind this effect as well, in which ATP would close K_ATP_ and H_2_O_2_ would active TRPM2 or other nonspecific calcium channels (NSCCs) [5], termed here as the “essentially synergic” channels (Figure 1B). Of course, both these mechanisms could also be in effect.

We also documented that the redox signaling upon GSIS is provided by elevations of cytosolic H_2_O_2_, whereas ROS in the mitochondrial matrix (both H_2_O_2_ and superoxide release) are diminished due to the enhanced operation of the redox shuttles upon GSIS [37] (see Section 4). Since the silencing of glucose-6-phosphate (G6P) dehydrogenase (G6PDH) in INS-1E cells abolished approximately half of the GSIS rate [4], one may conclude that a portion of cytosolic NADPH as the substrate for NOX4 is provided by G6PDH and downstream 6-phosphogluconate dehydrogenase within the PPP, while the second portion is generated due to the operation of redox shuttles, i.e., due to NADPH increasingly produced by isocitrate dehydrogenate 1 (IDH1) and malic enzyme 1 (ME1) in the cytosol upon glucose intake [37]. The essential role of PPP was emphasized elsewhere [38].

As a result, we can draw schematics for the revisited mechanism of the 1st GSIS phase (Figure 1 and Figure 2), while amplifying mechanisms based on the GLP-1 receptor (GLP1R) are also depicted in Figure 2. Elevated glucose metabolism, and specifically glycolysis, leads to an increased branching of the metabolic flux, particularly of G6P, toward PPP. Its enzymes, G6PDH and 6-phosphogluconate dehydrogenase, provide one of the NADPH sources for NOX4. PPP exhibits a rather low activity at low glucose, hence provides lower NADPH. Amplification of the cytosolic NADPH is also given by IDH1 and ME1 due to the elevated operation of the three redox shuttles described in Section 4. Since NOX4 was determined to be the only NADPH oxidase producing H_2_O_2_ directly [39,40], its elevated activity must lead to a profound increase in H_2_O_2_ release into the cytosolic compartment of β-cells. We have experimentally demonstrated in INS-1E cells and isolated islets that this indeed takes place, and that the excessive H_2_O_2_ release vanishes with NOX4 ablation [4].

In summary, the elevated H_2_O_2_, together with the concomitantly elevated ATP from the enhanced OXPHOS, is the only way to stimulate insulin secretion in a way that responds to the elevated glucose levels. As we discuss in detail in the Section 2.3, either ATP plus H_2_O_2_ are required for closing K_ATP_ [4] (Figure 1A); or ATP closes K_ATP_ and H_2_O_2_ activates the essential synergic channels (Figure 1B), such as NSCCs [5] or Cl^−^ channels [41]. These synergic channels shift the insufficient depolarization produced by the 100%-closed K_ATP_ population towards the depolarization of the plasma membrane over the approximately −50 mV threshold required for the opening of Ca_V_ [6] (Figure 3). In the latter case, there is no requirement for K_ATP_ to be redox-regulated, but it still could be.

Ca_V_ activation initiates action potential spikes, the descent part of which is governed by the opening of voltage-sensitive K^+^ channels (K_V_) (Section 2.3). The resulting pulsatile Ca^2+^ influx is the pacemaker for the created oscillations in cytosolic [Ca^2+^]_c_, which are the main effector for the complex machinery of IGV exocytosis. That is why IGV exocytosis is also pulsatile. This machinery is described below in Section 2.6 and in the Figure 2 legend.

Besides the 1st phase, there is also a 2nd phase of insulin secretion, given at least by the contribution of a distinct delayed kinetics of IGV release [42,43]. Nevertheless, since it has been recognized that the 2nd phase, which is manifested in pancreatic islets, does not exist in the β-cells isolated from islets [44,45,46], it was suggested to originate from the synchronization of electrical activity of the plasma membrane potential within the ensemble of cells in the islet, leading to synchronization of their cytosolic Ca^2+^ oscillations [47] and other events. A few percent of pacemaker-like β-cells are responsible for such synchronization in rodents [48,49]. These cells were termed hub cells. Nevertheless, the delayed kinetics of the IGV release plus intercellular synchronization act in parallel.

Mechanisms of both net GSIS 1st and 2nd phases were traditionally classified as triggering, since both involve Ca_V_ activation resulting in enhanced cytosolic [Ca^2+^]_c_ [49]. Moreover, the incretin-induced amplification mechanisms (typically induced by GLP-1 and GIP) are superimposed onto the canonical net GSIS mechanism. Thus incretins account for a large portion of the response to glucose in vivo (see Section 3). Multiple responses of β-cells to incretins mediated by incretin receptors further involve the additional incremental elevation of cytosolic Ca^2+^, such as from ER stores and the net facilitation of insulin granule vesicle docking, priming and exocytosis itself [23,26].

#### 2.1.2. Glucose Equilibration and Specific Metabolism in Pancreatic β-Cells

The consensus “canonical” mechanism of GSIS in the past has been an exclusive 100% dependence on the elevated OXPHOS upon glucose intake in pancreatic β-cells. Indeed, pancreatic β-cells were adapted by phylogenesis to serve as a perfect glucose sensor. This is already enabled by the specific isoforms of glucose transporters, GLUT2 in rodents and GLUT1 in humans, which provide equilibration of the plasma glucose concentration with the cytosolic glucose concentration in β-cells [50,51]. Furthermore, a β-cell specific isoform IV of hexokinase (termed glucokinase) is not feedback inhibited by its product G6P, which enables efficient unidirectional flux [52,53].

Now we know that this G6P flux is branched into glycolysis and PPP [38]. The participation of PPP in GSIS has been previously questioned, since PPP was found to account for 10% of glucose utilization and G6PDH was found to be inhibited by increasing glucose [54,55]. Nevertheless, pioneering metabolomics studies have already identified PPP intermediates as being strongly associated with GSIS [56]. Inhibition of the 2nd NADPH-producing enzyme, 6-phosphogluconic acid dehydrogenase within PPP, by 6-amino nicotinamide was reported to suppress GSIS in INS-1E cells after a chronic exposure [57] or in rats [58]. The inefficiency of the acute action of 6-aminonicotinamide was also reported [56]. Additionally, the PPP enzyme transaldolase was reported to be activated by glucose [59]. Moreover, patients with a G6PDH deficiency had the first phase of insulin release impaired [60].

The near absence of lactate dehydrogenase in β-cells and the inefficiency of pyruvate dehydrogenase kinases (PDK, which otherwise would block pyruvate dehydrogenase, PDH) then enables 100% of pyruvate and its equivalents (e.g., upon conversion by transaminases) to be utilized by OXPHOS. In pancreatic β-cells, PDK1 and PDK3 are “constitutively blocked” [61], and PDK2 does not phosphorylate the E1α subunit of PDH, hence it does not inhibit its activity. As a result, at low basal glucose, the β-cell PDH exhibited a relatively high activity, which is only inhibited by 22% at high glucose. Moreover, the activity of the matrix-localized complexes of PDH and 2-oxoglutarate dehydrogenase and the activity of NAD^+^-dependent isocitrate dehydrogenase 3 (IDH3) are enhanced by the incoming Ca^2+^ to mitochondria upon GSIS [49]. A major cataplerosis is ensured by the dominant citrate efflux from the mitochondrial matrix [62].

The predominant pyruvate utilization proceeds via the mitochondrial matrix PDH-complex followed by the complete Krebs cycle. However, a minor pyruvate flux, providing oxaloacetate anaplerosis, is given by the pyruvate carboxylase [63]. Its reaction is important for the pyruvate/malate redox shuttle and phosphoenolpyruvate shuttle described below. With glutamine present, aminotransferases such as cytosolic ALT1 and mitochondrial ALT2 (also termed glutamate pyruvate transaminases, GPT1 and GPT2 [64]) could catalyze the reversible conversion of pyruvate plus L-glutamate to 2-oxoglutarate (2OG) and L-alanine [65] during glutaminolysis. Mitochondrial 2OG would enter back into the Krebs cycle. However, it was suggested that the glutamate dehydrogenase instead synthesizes glutamate, which subsequently facilitates the exocytosis of insulin granules [66,67]. If this is the case, the reaction would act in the reverse direction, producing L-glutamate and contributing to the pyruvate pool.

#### 2.1.3. Two Phases of GSIS

There are two phases for the secretion of insulin responding to glucose in vivo [22,68,69], including GSIS in humans [70,71]. These two phases persist in isolated PIs. The K_ATP_-dependent mechanism (termed also “triggering”) and K_ATP_-independent mechanisms contribute to both phases [23]. Note that the K_ATP_-independent mechanisms still require the elevation of cytosolic Ca^2+^ [72]. Notably, it has been considered that the 2nd phase in vivo is independent of the extracellular glucose concentrations [73]. Mathematical models suggested that the 2nd phase of GSIS can be explained by the increased mobilization and priming of IGVs [74]. Thus also the cytosolic Ca^2+^ dynamics, which are even more complex, must contribute to the two phases. The issue of connectivity between β-cells within the pancreatic islets may also essentially contribute to the 2nd phase in rodents [45], but the mechanisms of how this is related to the delayed IGV kinetics are still to be elucidated. Additionally, note that human islets are morphologically different, and the connectivity issue has less importance, specifically for a high proportion of β-cell mass that is spread throughout the exocrine pancreas [75].

The first rapid and robust spike of insulin secretion occurs between 5 and 10 min after a bolus of glucose is administered in vivo or added to the isolated PIs. The explanation for this 1st phase of GSIS was based on how fast a fraction of existing IGVs can be released. The 1st phase was considered to be due to the predocked juxtaposed IGVs, located no more than 100–200 nm from the internal surface of the plasma membrane [76,77], and due to the newcomer IGVs, which may arrive within 50 ms and are not required to be predocked [78,79].

The 2nd phase was then suggested to depend on a functional recruitment of IGVs and mobilization of reserve IGVs, which, consequently with a delay, replace the previously instantly released pool. Additionally, the 2nd phase was previously thought to be realized by newcomer vesicles without priming at the plasma membrane, but delayed due to the passage through the filamentous actin (F-actin) cytoskeleton. F-actin was reported to be reorganized at the same time to allow the IGV passage [80,81,82]. Nevertheless, further research demonstrated that there is an amplification of insulin secretion, which is independent of microfilaments [81,83,84]. However, recently various cytoskeleton components have been demonstrated to play a more detailed and complex role in the IGV exocytosis [42,43], being more than just a barrier (see Section 2.6).

When experimentally induced by a 60 mM [K^+^] concentration to artificially depolarize the plasma membrane, the IGV exocytosis only affects those IGVs that are already docked to the plasma membrane, whereas glucose stimulation leads to a new recruitment of IGVs to the plasma membrane [85]. As a result, potentiation phenomena, collectively termed the amplification of insulin-secretion coupling, are involved in both phases of insulin secretion, but the “amplification” exhibits a delayed kinetics relatively to the “triggering”, which is instantaneous.

The 2nd GSIS phase can even last over 1 hr. Therefore, typically a higher amount of insulin is released in this phase. Experimentally, it became common practice to simulate the 2nd phase, either when K_ATP_ channels are kept open by diazoxide and KCl is added to depolarize the plasma membrane; or when glibenclamide (a sulfonylurea drug) permanently closes K_ATP_ independently of ATP. Both artificial manipulations reach the −50 mV depolarization threshold, hence enabling Ca_V_ opening and Ca^2+^ influx. Note, however, that the overall cell Ca^2+^ distribution between organelles such as mitochondria, ER and the cytosol is undoubtedly different (and so also the consequences for IGV exocytosis), when elevations of cytosolic Ca^2+^ originate from cell receptor (information) signaling. Note also that the role of glutamate was reconsidered, and instead of belonging to major factors influencing GSIS, glutamate is regarded as being essential for additional IGV exocytosis upon GSIS amplification by GLP-1 [67].

#### 2.1.4. Inconsistencies in Considering Exclusive GSIS Dependence on Elevated ATP in Pancreatic β-Cells

Several inconsistencies can be found, e.g., in INS-1E cells, which cannot be completely explained when considering the previous model for the 1st phase of GSIS, relying exclusively on the elevated ATP. At first, upon a complete block of the ATP-synthase with oligomycin, the closure of K_ATP_ is incomplete [4]. The ATP-synthase is the terminal OXPHOS enzyme, synthesizing ATP and releasing it into the mitochondrial matrix. Similarly, a predominant fraction of vestigial ATP-synthases with the missing subunit DAPIT in INS-1E cells enable GSIS, despite elevations of ATP in these cells being only 10% of those in non-transgenic cells [86]. These data indirectly support the requirement of H_2_O_2_.

Additionally, there is now an alternative interpretation of the reported experiments with long-chain acyl-CoAs. Long-chain acyl-CoAs were reported to bind to the Kir6.2 subunit of K_ATP_ [87], and therefore potently activate this channel [88,89]. Since GSIS was also found to be accompanied by a reduction in total cell acyl-CoAs and malonyl-CoA [62,90], it was hypothesized that this reduction facilitates K_ATP_ closure [62]. Nevertheless, an alternative explanation would be that β-oxidation contributing to the acyl-CoA decrease provides the redox signaling toward K_ATP_ and/or TRPM2, similarly as was described for KIC [4].

### 2.2. Ion Channels Participating in GSIS

#### 2.2.1. Plasma Membrane Potential and Ion Channels of Pancreatic β-Cells

The plasma membrane of the β-cell contains up to 60 channels of 16 ion channel families [91]. Several channels are also located on the membrane of IGVs. In general, a distinct pattern of channels exists in different species, various cultured model β-cells or even within individual cells of PIs [75]. The resting plasma membrane potential (*V*p^R^) is given by a greater concentration of K^+^ inside the β-cell (150 mM) than outside (5 mM) and predominantly by the activity of K^+^-channels. Hence, the plasma membrane permeability for K^+^ is higher than for other ions. The actually measured *V*p^R^ is −75 mV [92]. The closure of 100% of the K_ATP_ ensemble [91,93,94,95] was reported to lead to only an insufficient partial depolarization of the plasma membrane [6,91]. If the other channels did not contribute, the depolarization by the 100%-closed K_ATP_ ensemble would not be sufficient to activate Ca_V_ channels [96]. Thus, those “essential synergic” ion channels, notably redox-activated NSCCs, such as TRPM2, redox-activated NSCCs of other families [5], or even certain Cl^−^ channels [41], provide this essential enhancement of depolarization.

The resulting Ca^2+^ entry elevates the subplasmalemmal and cytosolic Ca^2+^ concentration and stimulates the Ca^2+^-dependent exocytosis of IGVs. However, Ca_V_ opening is intermittent with the opening of the voltage-dependent (K_V_) channels [97] or calcium-dependent K^+^-channels (K_Ca_) in humans. Thus K_V_ or K_Ca_ terminate Ca^2+^ entry. A simplified view describes that the single cycle of Ca_V_ opening followed by K_V_ opening determines each spike of the action potential firing. However, Na^+^-channels also help with creating upstrokes in 30% of the β-cell population [98]. The spike is repeated when K_ATP_ together with the essential synergy channels still maintain sufficient depolarization. When gaps exist between a series of spikes, such as at 10 mM glucose, a transient ATP consumption is responsible via sarco/ER-Ca^2+^-ATPase (SERCA) and plasmalemmal-Ca^2+^-ATPase (PMCA), i.e., ATPases removing Ca^2+^ [99]. In contrast, the involvement of ATP from cytosolic pyruvate kinases was indicated to add to the existing ATP pool during silent phases [100].

As for the fraction of K_ATP_ channels, the action potential firing is not induced until >93% of K_ATP_ channels are closed [101]. In human β-cells, this fraction is even higher [101]. So, in mice, the closure of the remaining 7% K_ATP_-population leads to the depolarization of the plasma membrane [102] in synergy with NSCCs or Cl^−^ channels. This is because the activity of the whole K_ATP_ channel population decreases almost exponentially with the increasing glucose concentration, the respective K_ATP_ current being already down to 50% at 2–3 mM glucose, and only 3% at 10 mM glucose [103]. The *V*p remains stable at 2–3 mM glucose. At about 7 mM glucose, the K_ATP_ channel current is so low that together with “synergic” channels, the attained depolarization leads to the action potential firing [91,92]. This situation is termed a supra-threshold depolarization. A nearly permanent firing exists at high >25 mM glucose [92]. Other channel phenomena cause the amplitude of permanent firing to become reduced by 15 mV after 3 min [75].

The voltage-dependent Ca^2+^-channels Ca_V_ in mouse pancreatic β-cells are predominantly of the L-type (Ca_L_) with a minor population of R-, N- and P/Q-type [104,105]. Ca_V_ channels contain four subunits, the pore forming the α1 subunit and auxiliary α2, β and γ subunit [104,105]. Several isoforms of the α1 subunit exist in pancreatic β-cells among species, such as Ca_V_1.2, Ca_V_1.3, Ca_V_2.1, Ca_V_2.2, Ca_V_2.3, Ca_V_3.1 and Ca_V_3.2, whereas the last two are found in human and rat β-cells [96]. They are responsible for different types of currents [104]. In mouse β-cells, Ca_V_1.2 and Ca_V_1.3 are responsible for 50%, Ca_V_2.1 for 15% and Ca_V_2.3 for 25% of the whole-cell Ca^2+^-current, which is activated at −50 mV [105]. Interestingly, Ca_V_2.3 contributes exclusively to the 2nd phase of GSIS [106]. The PKA-mediated phosphorylation of Ca_V_1.2 and Ca_V_1.3 enhances their activity [107].

#### 2.2.2. Detailed Sequence of Events in Plasma Membrane upon GSIS

An intermediate depolarization at 10 mM glucose was observed for mouse β-cells, which was reverted by a withdrawal of Ca^2+^ and Na^+^, evidencing the participation of other channels contributing to the depolarization (inward) flux, such as NSCCs [108]. Even an efflux of Cl^−^ was suggested to fulfill this role [109], including the opening of LRRC8/VRAC anion channels [41,110]. Among NSCCs, participation was also suggested for members 4 (TRPM4) and 5 (TRPM5) of the TRPM cation channel subfamily [111]. TRPM2 is redox-activated via the oxidation of Met191 [5] and activated by nicotinic acid dinucleotide phosphate (NAADP) [112], which is elevated upon GSIS [113]. Interestingly, TRPM2 was also reported to interact with peroxiredoxin 2, from which it can receive the redox signal [114,115]. Additionally, Ca^2+^-activated TRPM4 and TRPM5 channels, heat-activated TRPV1 (capsaicin receptor), and TRPV2 or TRPV4 belong to the important group of NSCCs expressed in β-cells [6]. Among them, the transient receptor potential canonical (TRPC) member 3 (TRPC3) channel was suggested to induce the inward shift in *V*p upon the activation of the GPR40 receptor [116].

As was noted above, the inward currents of certain levels by the opened NSCCs (or Cl^−^ channels) are essentially required for the induction of sufficient membrane depolarization in addition to the closing of 100% of the K_ATP_ ensemble [6]. This is already reflected by the fact that the resting potential *V*p^R^ of −75 to −70 mV is depolarized to some extent from the equilibrium *V*p^equi^ of −82 mV (calculated between 5 and 130 mM [K^+^]) (Figure 3). This shift is interpreted to exist due to NSCCs, since any Na^+^, Ca^2+^ and K^+^ ions can penetrate them [108]. Thus “synergic” channels provide a small background inward current that is unable to depolarize the plasma membrane at open K_ATP_, but able to do so with the predominantly closed K_ATP_-ensemble. The latter is possible since the NSCC-related conductance is then comparable to the small conductance provided by the remaining few % of open K_ATP_-channels (Figure 3). Without NSCCs, the established *V*p would only be equal to *V*p^equi^, so any shift to −50 mV required for Ca_V_ opening would not take place. As a result, the contribution by the basal opening of NSCCs is essential.

The same reasoning is also valid for anion channels, particularly Cl^−^ channels. Active Cl^−^ transport is provided in β-cells by SLC12A, SLC4A and SlC26A, which are able to set the cytosolic Cl^−^ concentration above thermodynamic equilibrium. Besides the consideration of GABA_A_, GABA_B_ and glycine receptor Cl^−^ channels being involved in the insulin secretion machinery, also volume-regulated anion channels (VRACs) were considered to be open at high glucose [75]. VRACs are heteromers of leucine-rich repeat containing 8 isoform A (LRRC8A) with other LLRC8 isoforms, forming anion channels [41]. The ablation of LRRC8 in mice led to delayed Ca^2+^ responses of β-cells to glucose and diminished GSIS in mice, demonstrating the modulatory role of LRRC8A/VRAC on membrane depolarization leading to Ca_L_ responses [110].

Intermittent Ca_V_ opening leads to *V*p oscillations, which may be further influenced by the Ca^2+^ efflux from the ER [91,92]. In isolated mouse PIs, glucose was reported to induce cytosolic Ca^2+^ oscillations, which were superimposed from fast (2–60 s periods) and slow (up to several min) Ca^2+^ oscillations [117]. They originate from *V*p oscillations of action potential firings and an interplay with Ca^2+^ efflux from ER [118], and evoke a pulsatile insulin secretion. The ER participation is provided by the phospholipase C (PLC) of the plasma membrane, which responds to the glucose-stimulated Ca^2+^ influx. PLC produces inositol 1,4,5-triphosphate (IP3), which opens the Ca^2+^ channel of the IP3 receptor (IP3R) of ER. Another ER Ca^2+^ channel, the ryanodine receptor (RyR) may also participate, being activated by ATP, fructose, long-chain acyl-CoAs and cyclic adenosine 5′-diphosphate ribose [117]. Additionally, the role of other channels permitting store-operated Ca^2+^ entry from the ER has been demonstrated, specifically the ternary complex of TRPC1/Orai1/STIM1 [6,119]. TRPC1 belongs to the TRPC family, which has a modest Ca^2+^ selectivity. TRPC1 interacts with Orai1 [120], and in such a functional complex, they are activated by STIM1 influx, the amplitude of Ca^2+^ oscillations and correlated with GSIS. The TRPC1 channels can be also recruited to the plasma membrane.

As was described above, the deactivation of Ca_V_ is achieved by the opening of K_V_ in rodents [97] or calcium-dependent (K_Ca_) K^+^-channels in humans. Among the former, tetrameric K_V_2.1 is the prevalent form in rodent β-cells. A delayed rectifier K^+^-current is induced at positive *V*p down to −30 mV [121]. The opening of K_V_2.1 channels repolarizes *V*p and thus closes Ca_V_ channels. The action potential spikes return to the plateau *V*p of −50 to −40 mV, the level of which is also adjusted by two-pore K^+^-channels TASK-1 and TALK-1 [122,123]. The ablation of K_V_2.1 indeed reduces Kv currents by 80% and prolongs the duration of the action potential, so more insulin is secreted. Mice with ablated K_V_2.1 possess a lower fasting glycemia but elevated insulin and reportedly improved GSIS [124]. In contrast, human β-cells use K_Ca_1.1 channels (also termed BK channels) for the repolarization of *V*p [92]. Note also that the downregulation of K_V_ was observed after islet incubation with high glucose for 24 h [125].

#### 2.2.3. Ablations of KIR6.2 and SUR1 Support a Central but not Exclusive Role of K_ATP_

PI β-cells of mice with ablated Kir6.2 (*Kcnj11* gene) (KIR6.2KO mice) did not exhibit typical K_ATP_ channel activity, but instead a higher resting *V*p and higher intracellular free [Ca^2+^]_c_ [126,127]. Glucose or tolbutamide (K_ATP_ inhibitor) only transiently elevated these resting free [Ca^2+^]_c_, and insulin secretion (GSIS) accounted for <10% of GSIS in wt mice PIs. Nevertheless, KIR6.2KO islets responded to 30 mM KCl by secreting insulin, i.e., when plasma membrane depolarization was achieved in this artificial way. KCl-induced depolarization is used by some researchers to model the 2nd GSIS phase. Incubations of KIR6.2KO islets with 10 mM glucose modulated Ca^2+^ oscillations, and any *V*p depolarization was K_ATP_-independent [127]. Moreover, despite this evident lack of K_ATP_ function, KIR6.2KO adult mice had a very mild impairment of glucose tolerance, recovering from the neonatal hypoglycemia and hyperinsulinemia [126]. All these results suggested a lack of the 1st phase of GSIS, but only the attenuated 2nd phase. They can be reinterpreted today by considering other channels substituting for the K_ATP_ function in membrane depolarization. Note also that in human pancreatic β-cells, Kir6.2 loss of function mutations lead to severe hypoglycemia and are accompanied by hyperinsulinemia in some forms [128].

Additionally, one can consider the phenomena observed in mice with ablated SUR1 (*Abcc8* gene) [129,130] as separately representing the so-called amplifying pathway of GSIS. SUR1 KO mice had an even milder impairment of glucose tolerance, but exhibit greater fasting hypoglycemia than KIR6.2 KO mice. Their β-cells exhibited a more depolarized *V*p. Moreover, insulin secretion was only potentiated by GLP-1 or GIP in wt PIs, but not in SUR1 KO PIs, whereas cAMP was elevated in both, reportedly in a PKA-independent way [131]. Since K_ATP_, and so also SUR1, reside in IGV membranes, SUR1 KO β-cells exhibited insufficient IGV docking and fusion with the plasma membrane [132].

#### 2.2.4. Structure of K_ATP_ and Behavior with Insulin Non-Stimulating vs. Stimulating Glucose

Despite the resolved structure and numerous mutagenesis studies of K_ATP_, no specific amino acid residues have been identified as being redox targets yet. The hetero-octameric K_ATP_ contains in total four external regulatory sulfonyl urea receptor 1 (SUR1, a product of *Abcc8* gene) subunits and four pore-forming subunits of the potassium inward rectifier Kir6.2 (*Kcnj11* gene) [133,134]. These four Kir6.2 subunits cluster in the middle of a structure with an 18 nm diameter and 13 nm height [135]. The cytoplasm-exposed part of Kir6.2 contains an ATP binding site, 2 nm below the membrane, which has been traditionally implicated in the channel closing, and an overlapping binding site for phosphatidylinositol 4,5-bisphosphate (PIP_2_). The binding of PIP_2_ stabilizes the open state. ATP binding to one of four ATP binding sites has already been reported to close the channel [136]. Moreover, the palmitoylation of Cys166 of Kir6.2 was found to enhance its sensitivity to PIP_2_ [137].

Pharmacologically, K_ATP_ is set in the open state by diazoxide, despite high ATP being present [138]. In contrast, sulfonylurea derivatives such as glibenclamide close K_ATP_, again independently of ATP, while binding to SUR1. Each of the four SUR1 subunits contain MgATP and MgADP binding sites. MgATP is hydrolyzed at nucleotide binding fold 1 (NBF1) to MgADP and then it activates K_ATP_ at NBF2, which is reflected by the ATP-sensitive increase in K^+^ conductance and consequent lower excitability, i.e., also lower sensitivity to ATP inhibition [136].

However, there is a discrepancy that is not yet fully resolved, concerning the drastically different sensitivities of K_ATP_ to ATP in vitro vs. in vivo. In inside-out patches used in the patch-clamp methodology, when the cytosolic side is exposed to the experimental medium and when so-called run-down is eliminated, as little as 5–15 μM ATP was able to close the channel [139]. There are much higher (mM) ATP concentrations in intact resting β-cells, albeit most ATP is bound with Mg^2+^. Despite the interaction of MgADP with SUR1 decreasing the sensitivity of the whole K_ATP_, this phenomenon cannot fully account for the above-mentioned discrepancy. Likewise, the requirement to close only the remaining 7% population of K_ATP_ does not encounter the typical S-shape inhibitory curve with an IC_50_ within the 10 μM range. Hence, there must either be endogenous K_ATP_ openers or the lack of H_2_O_2_ regulation and/or NSCC contribution could explain this phenomenon.

A variety of molecules were reported to be endogenous K_ATP_ openers. We already mentioned PIP_2_, which binds directly to KIR6.2 and decreases the ATP sensitivity of the channel. Upon the release of PIP_2_ from the binding site, the open probability is decreased [135,140,141]. Thus, for example, the extracellular activation of P2Y or muscarinic receptors by autocrine ATP (released together with insulin) decreases PIP_2_ via PLC activation.

#### 2.2.5. Possible Modulation of K_ATP_ by Kinases and Phosphatases in Pancreatic β-Cells

The phosphorylation of K_ATP_ was also thought to set the sensitivity of the ensemble of K_ATP_, so that transitions between the two distinct mM ATP concentrations, established by low (3–5 mM) vs. high glucose, will lead to the closing of the remaining fraction of the open K_ATP_ channels. Specifically, phosphorylation mediated by PKA could play a major role. Thr224 [142] and Ser372 were established as the candidate PKA phosphorylation sites. Their phosphorylation increases the open probability of K_ATP_ in insulin-secreting MIN6 cells [143]. This might hypothetically provide a closing mechanism that acts at higher ATP concentration or even requires H_2_O_2_.

The phosphorylation of K_ATP_ also increases the number of channels in the plasma membrane. Thr224 was also found to be phosphorylated by Ca^2+^/calmodulin-dependent kinase II (CaMKII) while interacting with β_IV_-spectrin [144]. In vivo, most likely autonomic innervations (maybe also paracrine stimulation) might provide sufficient PKA-mediated phosphorylation of K_ATP_. Hence, one should resolve how K_ATP_ function relates to phosphorylation in combination with the instantaneous modifications of sulfhydryl groups, which might substantiate the targets of the redox signaling. Thus, sulfhydryl groups of KIR6.2 and/or SUR2 might be affected either by the direct H_2_O_2_ diffusion or by peroxiredoxins and the redox relay system based on their action.

### 2.3. Possible Redox Regulation of Ion Channels

#### 2.3.1. Observed Redox Regulation of Ion Channels Participating in Insulin Secretion

Since the pioneering work of Ashcroft and colleagues [145], discovering the essential role of K_ATP_ in GSIS, only an indirect inhibition of K_ATP_ by H_2_O_2_ was observed in smooth muscle cells [146]. Otherwise, several candidates for redox-sensitive targets have been considered for pancreatic β-cells: (i) insulin granule exocytosis that might be directly induced by H_2_O_2_, independently of Ca_V_ triggering [97]. However, we excluded this possibility as being a major contributing factor, since in our experiments with INS-1E cells, independently of glucose, the ability of exogenous H_2_O_2_ to induce insulin secretion was only partially blocked by NOX4-siRNA, but it was fully blocked by the Ca_L_ blocker nimodipine [4]. As a result, despite the used H_2_O_2_ doses exceeding 100 μM, they did not directly stimulate the K_ATP_-independent exocytosis of insulin granules. (ii) Ca_V_ channels, when being coactivated by H_2_O_2_, might hypothetically serve as redox targets or (iii) H_2_O_2_ could hypothetically cause a direct or indirect inhibition of repolarizing K^+^-channels, such as K_V_ [147,148,149]. Finally, (iv) there could be a competition for NADPH between NOX4 and a hypothetical NADPH-activated K^+^-channel.

Nevertheless, using patch-clamped INS-1E cells in the cell-attached mode, we demonstrated a closure of K_ATP_ by H_2_O_2_ produced by NOX4 at high glucose, since in cells silenced for NOX4, even ATP resulting from the metabolism of high glucose was not able to close the channel [4]. According to our first hypothesis, any of the two K_ATP_ components, KIR6.2 or SUR1, should be redox-regulated so that H_2_O_2_ cooperatively with ATP closes the K_ATP_ channel (Figure 1A). The redox activation of TPRM2 depolarizing channels, which was already reported [150,151], could be an additional factor. An alternative hypothesis does not require the interaction of H_2_O_2_ with KIR6.2 or SUR1, but expects the essential activation of TRPM2. But this is not supported by the patch-clamp results [4].

Previously, the Ca^2+^-induced [97,152] or H_2_O_2_-induced exocytosis of insulin granules was reported upon the H_2_O_2_-activation of TRPM2 depolarizing channels [150,151,153]. Hypothetically, there may be an H_2_O_2_-activated TRPM2-dependent mechanism that provides the essential shift in depolarization to reach the threshold depolarization for Ca_V_ in synergy with the 100% K_ATP_ closure (Figure 1B). This mechanism of shifting depolarization was undoubtedly recognized for the GLP-1 potentiation of insulin secretion via the stimulation of TRPM2 by the GLP1R pathway [154]. Interestingly, it is not Cys residues but Met191 that is responsible for oxidation by H_2_O_2_ in the TRPM2 structure [5], and TRPM2 was also reported to possibly interact with peroxiredoxin 2 [114,115].

#### 2.3.2. Possible Redox-Target Residues of KIR6.2, SUR1 and TRPM2

In our model of Figure 4, we theoretically predicted H_2_O_2_-interacting cysteine residues of known structures of mouse KIR6.2 and hamster SUR1 (checking homologies in mouse vs. hamster sequences) and human TRPM2. We only emphasized those exposed to the cytosolic face of these channels. We based our search on recent findings in the field of redox proteomics by Chouchani’s group, who revealed interesting features of the potential ability of Cys to be oxidized [155]. Based on their analysis of 10 different mouse tissues, they proposed that redox-regulated cysteines seem to be tissue specific and that they exist in a local environment of surrounding amino acids, and that this environment affects the Cys reactivity towards redox modifications. At physiological pH, there is equilibrium between the thiolate/thiol group in Cys residues, and this equilibrium is sensitive to electrostatic modulation by nearby amino acids. The analysis revealed that the negatively charged amino acids aspartic and glutamic acid would favor the protonated Cys thiol. In contrast, the positive charge of arginine would stabilize the negatively charged thiolates, which are known to be much more susceptible to redox modifications [155,156].

Based on the above assumptions, we performed an analysis of potentially redox-regulated Cys within structures of mouse/hamster K_ATP_ and human TRPM2 channels. Hypothetically, the candidate target cysteines based on this analysis may be susceptible to interactions with H_2_O_2_, and thus may play a role in the above-described redox-activation of GSIS or in insulin secretion stimulated with BCKAs and partly FAs (see below). We searched for a Cys present in the accessible area of the intracellular domains of these proteins with favorable amino acid surroundings. According to the analysis, we found three Cys in the KIR6.2 subunit of the KATP channel (Cys42, Cys197 and Cys 344; Figure 4A), three Cys in the SUR1 subunit of the K_ATP_ channel (Cys1378, Cys1487 and Cys1491; Figure 4B) and two Cys in TRPM2 (Cys1250 and Cys1364, Figure 4C). All the suggested Cys share the feature of being in close proximity to an Arg that could increase their reactivity towards redox modification. At this point, it is hard to tell whether such a theoretical prediction has any validity in vivo, and extensive experimental evidence is needed to confirm these suggestions.

### 2.4. Fine Tuning of the Glucose-Sensitivity Range

#### 2.4.1. Inhibitory Factor IF1 as a Key Element Setting the Glucose-Sensitivity Range in Pancreatic β-Cells

Being originally termed an ATPase inhibitory factor 1 (IF1), this factor can now be called ATP-synthase inhibitory factor 1, emphasizing its experimentally evidenced ability to inhibit the synthesis of ATP in vivo [158,159]. Previously, it was thought that only the reverse mode of the ATP-synthase is inhibited by IF1, i.e., the mitochondrial ATPase reaction, pumping protons (H^+^) to the intracristal space at the expense of ATP hydrolysis to ADP. Such a mode is likely rarely established in primary cells, and the experiments demonstrating the ATPase mode were performed in yeast or cancer cells, in which this mode can coexist with a certain fraction of the ATP synthesis mode.

We recently demonstrated a surprising role of IF1, based on its ability to inhibit ATP synthesis in vivo, thereby adjusting the proper glucose concentration range for GSIS in rat pancreatic β-cells, INS-1E [158,159]. This conclusion was derived from the obtained results, demonstrating that IF1 silencing led to insulin secretion even at very low glucose approaching zero in INS-1E cells [158]. In contrast, IF1 overexpression inhibited GSIS in INS-1E cells [159].

Structurally, dimers of non-phosphorylated IF1 were reported to bridge the F_1_ moieties of neighboring dimers of the ATP-synthase, covering the crista rims in ordered rows. This was found within a tetrameric structure of the porcine ATP-synthase [160], and we can reasonably expect this to also occur in vivo. However, there is an important distinction in vivo. At most a very small minority of the dimers along the ATP-synthase row of dimers, which form cristae rims could be connected by these IF1-IF1 bridges.

If this was not the case, no ATP synthesis could take place, since IF1 bridging is inhibitory. In reality, IF1 binds at the interface between subunits β and α of the F_1_ moiety in proximity to subunit γ [161]. Hence, we could hypothetically expect a stoichiometry much lower than the saturating one, which is 1:3 for the IF1: subunit-β ratio. Moreover, IF1 was deactivated by mitochondrial PKA-mediated phosphorylation (see Section 2.5.4). Hypothetically, mitochondrial PKA might be involved in some mechanisms of amplification of insulin secretion. Cytosolic PKA, implicated in such amplification, could only participate if IF1 was phosphorylated before its import into the mitochondrial matrix. However, this speculation was not experimentally supported, as well as the speculation that IF1 expression might be even regulated in the translation stage [162]. In turn, experiments suggested a fast IF1 degradation by the factor IEX1 [163]. Consequently, we could expect a multifaceted fine-tuning of IF1 protein content in the matrix, including its phosphorylated fraction.

It is well established that the disruption/inhibition of the ATP-synthase activity induces a higher IMM electrical potential Δ*Ψ*_m_, slowing down the flow of electrons through the RC and thus allowing electron leakage to oxygen with concomitantly enhanced superoxide generation within complexes I and III [16]. Indeed, the upregulation of IF1 protein levels has been linked to increased superoxide formation and the activation of redox-sensitive transcription factors, such as NFκB or HIF1α, causing enhanced proliferation and survival in several cell types and tumors [164]. Consistently, in an INS-1E cell line overexpressing IF1, we observed a significantly higher endurance to stress induced by starvation or various inhibitors of the respiratory chain (Dlasková A., Kahancová A., unpublished). Moreover, it was suggested that IF1 prevents cytochrome c release during apoptosis via the stabilization of cristae structure due to enhanced ATP-synthase oligomerization [164]. In summary, it is likely that IF1 not only regulates the extent of insulin release, but it might also be critically involved in the regulation of β-cell number and survival under stressful conditions.

#### 2.4.2. Cristae Narrowing vs. Steepness of Glucose-Sensitivity Range in Pancreatic β-Cells

Due to the unsaturated and perhaps random IF1 bridging of the ATP-synthase dimers along their row forming the crista rim, ATP synthesis and GSIS may also depend on cristae morphology. We found a narrowing of mitochondrial cristae upon GSIS in INS-1E cells [165]. Proton pumps of RC supercomplexes, containing complexes I, III and IV, use the energy of electron transfer to pump H^+^ from the matrix space to the intracristal space. Crista junctions (outlets of the intracristal space) prevent the pumped H^+^ from diffusing out to the cytosol via the outer intermembrane space (space between the two cylindrical membranes of mitochondrial tubules, the outer membrane and inner boundary membrane). A local H^+^ coupling via the established protonmotive force Δp (Δp = Δ*Ψ*_m_ + ΔpH) thus occurs within the intracristal space [14]. As a result, any changes in the volume of the intracristal space will affect the coupling efficiency, i.e., the relationship between the Δp and the H^+^ backflow through the ATP-synthase F_O_ sector back to the matrix.

It remains to be determined whether the observed cristae narrowing upon GSIS also diminishes the volume of the intracristal space. In the positive case, one can predict that crista narrowing in the mitochondria of pancreatic β-cells strengthens the proton coupling. This would contribute to the steepness of the dose-response relationship of glucose concentration vs. insulin release. Other complex relationships may exist between the IF1 stoichiometry relative to the ATP-synthase, the cristae morphology, and other cristae-shaping proteins such as OPA1 and Mic10.

#### 2.4.3. Responses of the Mitochondrial Matrix Calcium during GSIS

Mitochondrial Ca^2+^ was also recognized to be a significant regulator [166] or participant in the 2nd GSIS phase and in the potentiation of GSIS by GLP-1 [49]. The pioneering work of Denton and colleagues demonstrated the existence of the activation of mitochondrial dehydrogenases by the matrix intake of Ca^2+^ [167], which, despite the contradictory findings of some researchers [168], could take place upon GSIS [99,169,170,171] or upon GSIS amplification by GLP-1 [172,173]. The mitochondrial matrix concentration [Ca^2+^]_m_ is regulated by the Ca^2+^ influx, mediated by the mitochondrial calcium uniporter (MCU) [174], which is balanced by the Ca^2+^ efflux. The latter is provided by the mitochondrial Ca^2+^/Na^+^ antiporter (NCLX) [175], the fluxes of which are driven by the mitochondrial Na^+^/H^+^ antiporter (NHE6), at the expense of the protonmotive force. In some cell types, the existence of the mitochondrial Ca^2+^/H^+^ antiporter was also reported. Its expression in β-cells is ascribed to the LETM1 [176].

When MCU-mediated Ca^2+^ influx exceeds the Ca^2+^ efflux, the cytosolic transient Ca^2+^ elevations caused by the Ca_V_ channels are relayed into the mitochondrial matrix [177]. A sudden intake of glucose in primary β-cells induces the primary Ca_V_-dependent [Ca^2+^]_c_ oscillations. Those are relayed to somewhat delayed steady-state increases in mitochondrial [Ca^2+^]_m_, which reach saturation, i.e., maximum values after a certain time [99,178]. The [Ca^2+^]_m_ oscillations, responding to those ongoing in the cytosol, are then superimposed onto the linearly increasing [Ca^2+^]_m_ up to saturation. When bursting occurred at 10 mM glucose, the higher frequency of action potential spikes within a burst led to a higher amplitude of the [Ca^2+^]_m_ increase, i.e., higher saturating values [178]. These changes induced a biphasic increase in the ATP/ADP ratio, with its 2nd part after 5 min [99,177,178].

MCU-deficient β-cells did not exhibit the 2nd part of the increase in ATP/ADP [99]. Note that this does not necessarily correspond to the 2nd GSIS phase. When the artificial overexpression of the Ca^2+^ binding protein S100G in the mitochondrial matrix of INS-1E cells allowed declines in the matrix Ca^2+^ independently of the cytosolic Ca^2+^, the glucose-stimulated NAD(P)H formation was prevented, and increases in mitochondrial respiration and OXPHOS [179]. Thus, glucose-induced [Ca^2+^]_m_ elevations reached up to 880 nM in wt cells, whereas only 530 nM upon the S100G overexpression, which attenuated GSIS. In contrast, the overexpression of S100G in islet β-cells did not decrease the 1st phase of GSIS, but attenuated the 2nd phase.

### 2.5. Amplifying Mechanisms

#### 2.5.1. Receptor Mediated Amplification

G protein-coupled receptors provide a wide range of cell responses that are mutually interrelated. They activate heterotrimeric G proteins, which typically regulate the production of second messengers or signals via other proteins. Thus, the G protein Gαs increases the generation of cyclic AMP (cAMP), whereas Gαi/o decreases it [180,181,182]. Gαq/11 initiates the PLC-mediated hydrolysis of phosphatidylinositol 4,5-bisphosphate (PIP_2_) into diacylglycerol (DAG) and IP3 [183,184]. Gα12/13 promotes the protein RhoA for remodeling the cytoskeleton [185]. The class of proteins termed β-arrestins initiate signaling via the proximal MAP kinase, IκB and Akt pathways [186]. Downstream pathways are discussed below, which predominantly lead to either (i) modulation of the plasma membrane channels, typically Ca_V_, K_ATP_ and K_V_, to provide more intensive insulin secretion; (ii) stimulation of the surplus Ca^2+^ influx to the cytosol from ER or other stores and (iii) influencing the kinetics of the IGV in its docking, priming and fusion with the plasma membrane. Note, that only (iii) mechanisms could be independent of Ca_V_, if other factors provided sufficient [Ca^2+^]c increase. Most likely, however, there is usually a synergy between all of these mechanisms (i) to (iii). The general role of second messengers in pancreatic β-cells is described below, while the participation of each pathway in insulin secretion stimulated by various secretagogues is described in later sections.

#### 2.5.2. PKA Pathway in Pancreatic β-cells

The activation of Gαs increases the activity of plasma membrane transmembrane adenylate cyclases (tmAC) producing cAMP from ATP [181,182]. A number of phosphodiesterases of 11 families then degrade cAMP (some also or exclusively cGMP). cAMP is a universal 2nd messenger providing a vast number of diverse physiological functions, being involved in metabolism, differentiation, synaptic transmission, ion channel activity, growth and development or in cellular level in cell migration, differentiation, proliferation and apoptosis. There are also soluble adenylate cyclases (sACs), whose reaction is potentiated by Ca^2+^ and bicarbonate. One such sAC is also located in the mitochondrial matrix of pancreatic β-cells [187]. The major mediator of cAMP effects is the cAMP-dependent PKA [188]. Note that PKA is ubiquitous in every mammalian cell and acts in the cell cytosol, nucleus and evidence has also been gathered that it acts in mitochondria, either tethered to the outer mitochondrial membrane (OMM) or located in the matrix [189,190].

PKA is a heterotetramer of two distinct catalytic subunits and a dimer of a regulatory subunit. Three distinct catalytic (Cα, Cβ, and Cγ) and four isoforms of the regulatory subunit exists (RIα, RIβ, RIIα and RIIβ), which bind cAMP [188]. After cAMP binding, the catalytic subunits dissociate and phosphorylate a vast number of proteins. PKA is tethered by A-kinase anchoring proteins (AKAPs), localizing it to specific sites in the cell [189], e.g., externally to the OMM from the cytosolic side. The PKA is thus positioned in close proximity to the dedicated protein substrates. Phosphodiesterases (PDEs) degrade cAMP and contribute to the termination of signaling. For example, in β-cells PDE3B attenuates the potentiation of insulin secretion by GLP-1 [191].

In pancreatic β-cells, the PKA pathway is involved in the signaling of incretin (typically GLP-1 and GIP) receptors [180], and makes a minor contribution to signaling from metabotropic receptors, such as GPR40, sensing fatty acids [184]. PKA typically amplifies the Ca^2+^-dependent exocytosis of insulin granules. The core pathway involves the PKA-mediated phosphorylation and hence activation of the Ca_V_ β2-subunit, in concert with the phosphorylation of K_ATP_, increasing the ATP concentration range required for its closure [192]. In addition, PKA inhibits Kv channels, which otherwise terminate plasma membrane depolarization, hence this prolongs the already more intensive Ca^2+^ influx via phosphorylated Ca_V_ channels and hence prolongs/intensifies the exocytosis of IGVs [193].

An exocytosis modulating the protein snapin is another important target of PKA. The phosphorylation of snapin enables its interaction with the other proteins of the IGV, thus potentiating the 1st GSIS phase [194]. Snapin participates in tethering IGVs to the plasma membrane by coiled–coil interaction with a lipid-anchored protein, the synaptosomal nerve-associated protein 25 (SNAP-25) [195].

#### 2.5.3. EPAC2 Pathway in Pancreatic β-Cells

The PKA pathway provides approximately 50% of the responses to cAMP in β-cells [196]. The parallel pathway of enhanced signaling via exchange proteins directly activated by cAMP 2 (EPAC2) enables the remaining portion [197,198,199]. Due to its guanine nucleotide exchange activity, the EPAC2 protein provides Ca^2+^-induced Ca^2+^release from the ER via RyR [200], although the abundance of RyR in β-cells has been questioned [201]. Nevertheless, the RNAseq approach indicated the existence of RyR transcript in mouse and human β-cells. The Ca^2+^release from ER via RyR can only happen upon abundant glucose intake, leading to the primary Ca_V_ opening [202]. The primary increase in [Ca^2+^]_c_ then reaches a threshold, at or above which the RyR Ca^2+^ channel would be open. Note also that sustained Ca_V_ opening partially refills the ER Ca^2+^ stores, hence counteracting the above effect. The most important effect of EPAC2A is the direct activation of TRPM2, essential for K_ATP_-triggered GSIS, i.e., providing an NSCC-mediated depolarization shift of up to −50 mV [154]. The redox status of TRPM2 under these conditions is not known.

The EPAC pathway also affects IGV proteins and therefore facilitates IGV exocytosis. Indeed, the cAMP potentiation of insulin secretion via distinct EPAC2 pathways involves the regulation of several IGV-associated proteins. The role of Rim2α GTPase was elucidated [203,204]. Rim2α is located on the inner plasma membrane surface and on the membranes of insulin granules. Thus, Rim2α represents a scaffold that enables the regulated exocytosis of insulin granules [205]. While interacting with another GTPase Rab3A, located on insulin granules, the Rim2α–Rab3A complex enables the docking of IGVs (Figure 2). Priming is then initiated when Rim2α interacts with mammalian uncoordinated homology 13-1 (Munc13-1), which subsequently opens syntaxin 1 to interact with the IGVs. This is followed by fusion with the plasma membrane. Rim2α double mutants K136E/K138E locked Rab3A to Rim2α so that priming cannot proceed and syntaxin 1 remains closed.

Interestingly, EPAC2 also binds to the nucleotide binding domain-1 of the SUR1 K_ATP_ subunit. The EPAC2 release from SUR1 is then provided by cAMP [25]. Hypothetically, this locally released EPAC2 induces the release of Rim2α from the α1.2 Ca_L_ subunit [180]. Ongoing Ca^2+^ influx facilitates EPAC2 binding to Rim2α and trimers of these two proteins with another Ca^2+^ sensor, Piccolo. The trimers interact with Rab3A and thus enable IGV exocytosis.

#### 2.5.4. Mitochondrial PKA Pathways in Pancreatic β-cells

It has been found that PKA phosphorylates numerous mitochondrial proteins, including proteins of the mitochondrial matrix [206,207]. Therefore, we must consider PKA to be a parallel effector that also involves cAMP signaling in the mitochondrial matrix of pancreatic β-cells [189,190]. Suitable protein residues exposed to the cytosolic face of the OMM are phosphorylated by the PKA residing on the OMM (PKA_OMM_). The PKA attachment is mediated by the OMM-anchored AKAP. In this way, the fission/fusion dynamics of the mitochondrial network are regulated, besides the protection of apoptosis. As a result of this OMM PKA signaling, typically mitochondrial tubule elongation is promoted within the mitochondrial network in the cell, with concomitantly increased mitochondrial membrane potential, besides numerous other phenomena. Additionally, the complex IV COXIV-1 subunit (which prevents its inhibition by ATP) is phosphorylated by PKA from the intracristal or the outer intermembrane space (IMS_out_), i.e., the space between the OMM and the inner boundary membrane (IBM) [208]. The IBM is an unfolded IMM part forming a cylindrical IBM-IMS_out_ –OMM sandwich.

From the unambiguous fact that only the non-phosphorylated IF1 inhibits the ATP-synthase, one can derive a hypothesis that expects mtPKA localized in the mitochondrial matrix to activate the ATP-synthase by providing phosphorylation of the major ensemble of IF1 protein. This was already verified in INS-1E cells [158,159]. The question is whether such regulation can be acute, i.e., whether it is in effect during the 1st or 2nd GSIS phase. Evidence has accumulated for the existence of the PKA pathway in the mitochondrial matrix. All the necessary components were identified to be localized in the matrix, including sAC, PDE2A2 [209] and also the matrix PKA (mtPKA) [210], albeit its existence is still controversial. A pharmacological activator of GPRs, forskolin, was reported to induce the phosphorylation of matrix proteins. The ATPase inhibitory factor IF1 was reported to be phosphorylated. However, there is still disagreement over whether this takes place in the mitochondrial matrix [164] or whether IF1 is phosphorylated by the cytosolic or OMM-residing PKA prior to its import to the matrix [211]. There was also a consensus that cAMP cannot freely diffuse to the matrix [209]. Thus, cAMP located in the mitochondrial matrix represents an independent pool, which may therefore provide an independent regulation [212,213].

The most likely source of matrix cAMP is the matrix sAC activated by bicarbonate [187] and Ca^2+^, though it was reported that a matrix cAMP increase does not activate mtPKA [214]. Since CO_2_ is increasingly released when the Krebs cycle turnover increases upon the metabolic stimulation of insulin secretion, we can hypothesize that the elevation of cAMP is independent of the cytosolic cAMP signaling in pancreatic β-cells. Additionally, the basic [Ca^2+^]_c_ increase due to Ca_V_ opening and its relaying to a [Ca^2+^]_m_ increase should activate sAC. The main effector is most likely the matrix-localized mtPKA [215], although nothing is yet known about how PKA is translocated into the mitochondrial matrix. The matrix localization of PKA was clearly demonstrated in *Drosophila* [210].

In any case, OXPHOS is facilitated in the mitochondria of numerous tissues, stemming from the phosphorylation of complex I NDUFS4 subunit (facilitating its Hsp70-mediated import) and IF1 (enhancing ATP synthesis by disabling the inhibitory binding of phosphorylated IF1 dimers to the ATP synthase; see above) [211] and there is no reason for this to not also be the case in pancreatic β-cells. Indeed, this was confirmed [158,159]. The observed release of the PKA catalytic subunits by the increased ROS is also noteworthy [216,217].

We may hypothesize that the metabolic branch of the amplification of insulin secretion originates from the elevated mitochondrial matrix cAMP and activated mtPKA, at least due to the increased CO_2_ output in the Krebs cycle and increased Ca^2+^ uptake into the mitochondrial matrix. Note that CO_2_ is converted to bicarbonate by the carbonic anhydrase [218]. Thus mtPKA could accompany the cytosolic PKA signaling initiated by the GPR40 and GLPR or GIPR receptors. In other words, mtPKA signaling should contribute to the amplification of insulin secretion by FAs or incretins while phosphorylating at least IF1, and probably also complex I.

We should also note that a partial suppression of ATP synthesis is also projected into the RC-produced superoxide. Indeed, RC proton pumps work faster at faster ATP-synthesis, given by the H^+^ backflow to the matrix. When the latter is retarded, such as due to the binding of non-phosphorylated IF1 dimers, then RC proton pumps also retard their H^+^ flux against Δp, while RC electron transfer also slows down. This leads to a higher superoxide formation, which would otherwise decrease upon the increasing ATP-synthesis. In this way, the mtPKA-IF1 pathway may slightly contribute to the redox signaling upon GSIS, adding to the H_2_O_2_ fraction resulting from surplus superoxide that is formed and converted upon slightly retarded ATP synthesis.

#### 2.5.5. Gαq/11-IP3 and Gαq/11-DAG-PKC Pathways

The G protein Gαq/11 initiates signaling through the phospholipase C (PLC)-mediated hydrolysis of phosphatidylinositol 4,5-bisphosphate into DAG and IP3 [183]. The most important contribution of DAG is the facilitation of the threshold plasma membrane depolarization, via the protein kinases C- (PKC-) mediated phosphorylation of TRPM4 and TRPM5, leading to their activation. These channels belong to the essential synergic channels for K_ATP_ [219]. PKCs hence probably contribute to the required essential *V*p shift, otherwise provided by TRPM2. One of the effectors of IP3 is the IP3 receptor (IP3R; subtypes IP3R1, IP3R2 and IP3R3), forming another important Ca^2+^ channel residing on the ER membrane in β-cells [220]. Similarly to the EPAC2-RyR route of Ca^2+^ release from the ER, the opening of this channel amplifies the primary Ca_V_-mediated Ca^2+^ signaling for insulin release.

PKCs isoenzymes, which belong to the family of serine/threonine kinases, are activated by DAG, which is their main effector [221,222,223,224]. PKCs are important for a broad range of cellular processes [225]. Pancreatic β-cells express members of both conventional PKCs (cPKCs), which are activated by DAG and Ca^2+^, and “novel” PKCs (nPKCs), which are activated by DAG, but not by Ca^2+^. This enables nPKCs to be activated by a general PLC-mediated hydrolysis of lipids other than PIP_2_. The activation of cPKCs and nPKCs typically involves their translocation to the plasma membrane [223]. DAG-stimulated nPKCs reportedly also migrate to the Golgi apparatus and mitochondria (PKCδ to nucleus) [225]. PKC colocalization with the OMM has been reported [226]. The translocation dynamics of various fluorescence-tagged PKC isoforms together with DAG dynamics, and the subplasma membrane [Ca^2+^], were analyzed in insulin-secreting β-cells using TIRF [223]. The results revealed that insulin secretagogues induce transient DAG microdomains, which rapidly recruit both cPKCs (specifically isoforms PKCα, PKCβI and PKCβII) and nPKCs (isoforms PKCδ, PKCε and PKCη), emphasizing the role of DAG in the rapid kinetic control of PKC-mediated phosphorylation. The pharmacological stimulation of GPR40 also activated PKCε at a substimulatory (glucose), while in addition to PKCε, PKCα was also activated at high insulin-stimulating (glucose) [224]. nPKCs were found to affect cytoskeleton dynamics in a way that eases IGV exocytosis [227]. Besides their canonical plasma membrane effects, PKC [227] and downstream ERK1/2 signaling stimulate OXPHOS, and hence mitochondrial ATP synthesis [227]. Thus, PKC signaling was claimed to target mitochondria [225,228], since a range of nPKC inhibitors prevented phorbol-ester-induced respiration in INS-1E cells and human PIs [227]. However, the elevated OXPHOS was affected indirectly via c-Raf–MEK–ERK1/2 signaling. Thus, the existence of matrix-located PKC in pancreatic β-cells is not clear.

### 2.6. Biology of Insulin Granule Vesicles

#### 2.6.1. Biogenesis of Insulin Granule Vesicles

About 10,000 granules are present in rodent β-cells [229]. Besides insulin hexamers crystallizing with zinc ions Zn^2+^, IGVs contain C-peptide, ATP, glutamate, GABA, ghrelin and islet amyloid polypeptide (IAPP, termed also amylin). All these factors exhibit an autocrine function. IGVs have their own membrane, ion channels and protein machinery, which is not completely understood [75]. Thus, IGVs represent self-contained organelles that are 200–400 nm in diameter, apparent in electron micrographs as having a dense core with a white halo in rodents [77,230]. IGVs contain a high content of total Ca^2+^ (that would correspond to 120 mM), most of which is bound [231]. The free Ca^2+^ concentration may correspond to 40 μM [232].

The biogenesis of IGVs is initiated by the signal of high glucose, which serves for the instant maintenance of replenished IGVs [77,233]. Thus, the translation of preproinsulin is dependent on glucose [233], which is followed by the processing of proinsulin in the ER [234] and the transition to the Golgi apparatus and its trans Golgi network, from which nascent IGVs bud [235]. In these nascent IGVs, 8.93 kDa proinsulin is cleaved by prohormone convertases (losing a C-peptide), and insulin is matured as a 51-residue 5.808 kDa protein and crystallized into hexamers in association with Zn^2+^ [230,236] and bound Ca^2+^ [49]. About 8–9 fg (1.7 attomoles) of insulin gives a 100 mM concentration in a single IGV. All the insulin accounts for 5–10% of β-cell proteins [75]. Additionally, microtubules were recently implicated in this process, with the involvement of the EPAC2 pathway increasing the formation of such microtubules derived from the Golgi apparatus [237]. The process was found to support both phases of insulin secretion. The nascent IGVs are transformed into immature IGVs (less strongly stained with a thin halo or without halos) coated with clathrin, while acidification excludes the clathrin-coated membrane parts, which produces the mature IGVs [42]. The transfer of IGVs towards the plasma membrane is achieved through sliding along microtubules.

#### 2.6.2. Major Proteins of IGVs

Proteomic analysis indicated numerous proteins contained in IGVs [238]. Besides insulin, numerous enzymes were found inside these vesicles, such as chromogranin-A, carboxypeptidase E, ectonucleotide pyrophosphatase/phosphodiesterase family member 2, fibronectin, stanniocalcin-1, several neuroendocrine convertases, sulfhydryl oxidase, 2′,3′-cyclic-nucleotide 3′-phosphodiesterase, carboxypeptidase N, β-galactosidase and others. Notable proteins found attached to or within the IGV membranes were synaptotagmin-like protein 4, Rab37, Rab1α, vesicle-associated membrane protein 3 (VAMP3), syntaxin-12, syntaxin-5, V-type H^+^ATPase, receptor-type tyrosine-protein phosphatase N2, SLC12A9, carboxypeptidase D and transferin receptor protein 1, among numerous others. Several vesicle-associated membrane proteins (VAMP), notably synaptobrevin/VAMP2, reside on the IGV surface [238]. Synaptobrevins and similar IGV proteins are termed v-SNARE to distinguish them from the target SNARE proteins of the plasma membrane. As described below, these soluble N-ethylmaleimide-sensitive factor (NSF) attachment protein receptor (SNARE) proteins are key proteins that provide the exocytosis of IGVs by allowing the fusion of two membranes.

IGV membranes contain zinc transporter member 8 (ZnT8/SLC30A8), which allows a high accumulation of Zn^2+^ in the IGV lumen to promote the insulin crystalline form [239]. During IGV exocytosis, thus also Zn^2+^ is released together with C-peptide in addition to the crystalline insulin, ATP, GABA, glutamate, ghrelin, IAPP/amylin and other components. Surprisingly, channels in the IGV membrane involve the K_ATP_ channels [240,241], ClC3 channels (see below), TRPV5 [242], RyR [243] and probably also IP3R [244]. The latter three channel types serve for Ca^2+^ mobilization from IGVs, thus contributing to Ca^2+^ oscillation dynamics. The role of K_ATP_ in IGVs is less clear. Some clues could be provided by the interaction of the small cytoplasmic phospholipid binging protein α-synuclein with the K_ATP_ of IGVs, which inhibits the basal insulin release, but not GSIS [241].

Overall changes in IGV interiors are given by the V-type H^+^ATPase, which pumps H^+^ into the IGV lumen, hence establishing protonmotive force with the electrical potential Δ*Φ*_IGV_ and ΔpH_IGV_ components. The concomitant Cl^−^ influx, mediated by the ClC3 channel, contributes to ΔpH_IGV_. Similarly, Δ*Φ*_IGV_ and ΔpH_IGV_ drive all transport processes across the IGV membrane. The ATPase H^+^-transporting lysosomal accessory protein 2 (ATP6ap2) is required for the assembly of V-H^+^ATPase in IGVs. Reportedly, ATP6ap2 interacts with GLP1R, while its silencing decreased both GSIS and its potentiation by GLP-1 in INS-1E cells [245].

#### 2.6.3. Mechanism of IGV Exocytosis

A small portion of insulin was found to be released without being contained in IGVs [246]. IGV release is pulsatile, in accord with the Ca^2+^ oscillations, and not continuous [247]. When accounting for averaged rates, a fraction of 0.14% min^−1^ of IGVs was found to be released in the 1st phase and 0.05% min^−1^ in the 2nd phase of GSIS [42,68,230]. The latter would account for 3% of IGVs after 60 min from those existing prior to the transition to a high glucose. In a snapshot, up to 1% of IGVs were characterized as readily released, 7% IGVs were found to be docked to the plasma membrane, 15% were found near the plasma membrane, but not docked and the remaining 77% of IGVs represent a large reserve pool [79]. Docking was found to not be essential, but produced a delay in fusion with the plasma membrane [248].

The major Ca^2+^ signal preceding insulin secretion allows Ca^2+^ binding to the C2 domains of complexin-1 and C2A and C2B domains of certain synaptogamins [249,250] (Figure 5). So-called priming of IGVs follows and involves a conformation change from the closed to the open form of the triple-α-helical Habc N-terminal domain of syntaxin-1 (Figure 5, transition from the stage 1 to stage 2). The mammalian uncoordinated-18 protein (Munc18-1) binds to the closed form, preventing any participation in the SNARE complex [251]. Additionally, Munc13-1 facilitates the syntaxin transition in concert with some other proteins [249]. In parallel with synaptogamin activation by Ca^2+^, the plasma membrane proteins syntaxin-1, SNAP-25 and other target-SNARE proteins attract IGVs via synaptobrevins and other VAMP proteins, while forming a coiled-coil quaternary structure [252]. The inherent part of such a complex is given by the Ca^2+^-bound protein, termed complexin-1 (Figure 5, stage 2). These events build a four-component bundle coiled-coil SNARE complex.

The resulting SNARE core complex relocates closer to the plasma membrane so that the IGV membrane is moved into its proximity, and thus facilitates establishment of the so-called fusion stalk. Further zippering of coiled-coil structures allows the fusion of a larger part of the IGV membrane with the plasma membrane, until a fusion pore is formed [250,252]. A further expansion of the fusion pore is promoted due to the *cis* to *trans* conformation of the SNARE complex and, simultaneously, the entire IGV lumen content is relocated to the cell exterior (to islet capillaries) (Figure 5, stage 3). Note that also the PKA-phosphorylated Snapin aids the IGV tethering to the plasma membrane by coiled coil interaction with a protein SNAP-25, which is anchored to the plasma membrane by a palmitoyl chain interacting with the cysteines of the random coil linker region joining the Qb and Qc domains [195].

As was mentioned above, small scaffold GTPases Rim2α and Rab3A, in a complex, interact with Munc13-1, which subsequently opens the syntaxin. Being GTPases, these proteins are activated by the cAMP-dependent EPAC2A pathway. In the end, the NEM sensitive factor (NSF), being a homo-hexameric AAA-type ATPase, disrupts the SNARE complexes at the expense of ATP hydrolysis (Figure 5, stage 4), so the exocytosis is completed. Palmitoylated SNAP-25 dissociates automatically from the membrane upon fusion.

#### 2.6.4. Glutamate Promotion of Insulin Granule Vesicles Exocytosis

The originally described key role of glutamate in relation to GSIS [253,254,255] was questioned by subsequent experiments [256,257]. In turn, total internal reflection fluorescence microscopy (TIRF) studies demonstrated that membrane-permeant dimethyl-glutamate amplified both phases of GSIS with respect to the frequency of IGV merging with the plasma membrane [258]. All previous controversial findings thus probably reflect the specific uptake of glutamate into IGVs driven by ΔpH_IGV_, established by the V-H^+^ATPase [259]. Anion influx, namely Cl^−^ influx by ClC3, helps to build ΔpH_IGV_.

The glutamate concentration inside the IGV lumen is maintained by the EAAT2-mediated efflux, while being balanced by the uptake into the lumen, mediated by vesicular glutamate transporters VGLUT1, VGLUT2 and VGLUT3. All these reside in the vesicular membranes, being differentially expressed in various cultured model β-cells and islet β-cells [260]. Their ablation did not affect GSIS, but reduced the incretin-induced amplification of GSIS [258]. The impaired glutamate vesicular transport thus decreases overall insulin secretion. Glutamate dependence was also judged from experiments with ablation of the glutamine transporter of the plasma membrane, sodium-coupled neutral amino acid transporter 5 (SNAT5). The insufficient glutamine uptake and concomitant glutaminase reaction, converting glutamine to glutamate, reportedly lead to the reduced GLP-1 amplification of GSIS [261].

## 3. Incretins: GLP-1 and GIP

### 3.1. Incretin Potentiation of GSIS In Vivo

Glucagon-like peptide 1 (GLP-1) and gastric inhibitory polypeptide (GIP) have a prominent impact among other peptides belonging to the incretins. Oral glucose administration provides a much higher insulin secretion response compared to parenteral administration [262]. The potentiation surplus can be about equally ascribed to GLP-1 and GIP [263]. This is reflected, for example, by the diminished insulin secretion response to oral glucose in GLP-1 knockout mice [264,265], which decreased even more when double knockout (GLP-1 plus GIP) mice were tested [265].

Incretins (GLP-1 or GIP) massively amplify GSIS by signaling via Gαs proteins, tmAC, cAMP and subsequently via both PKA-dependent and EPAC2A-dependent pathways. As described above, the phosphorylation by PKA increases the sensing range of K_ATP_ to physiological mM ATP, inhibits K_V_, activates Ca_V_ [266], the GLUT transporter and snapin [194,195]. Simultaneously, the EPAC2A pathway directly activates TRPM2, essential for K_ATP_-triggered GSIS, i.e., ensuring the NSCC-mediated depolarization shift up to −50 mV [154]. Since this shift is also essential for the net non-amplified GSIS, the cause(s) of the additional triggering and/or increased time of bursts for Ca_V_ opening and/or more inhibited K_V_ (prolonging spikes) needs to be resolved. Collectively, these events lead to the increased integral [Ca^2+^]c that determines the surplus insulin secretion_._

EPAC2A also directly activates Rim2α [205,253], thus facilitating SNARE complex formation, relocation (IGV priming) and the subsequent *cis* to *trans* conformation change of the major SNARE complex (see above). Moreover, the EPAC2A pathway via Rap2-calmodulin kinase II (CaMKII) activates RyR, ensuring Ca^2+^ efflux from the ER, preceded by Ca_V_ opening, thus again contributing to the amplification of IGV exocytosis [267]. When there is biased GLP1R (GIPR) signaling via Gαq/11, it activates PLC and either IP3-IP3R-mediated or PLC plus STIM1-Ora1-TRPC1-mediated Ca^2+^efflux from the ER (or TRPC1 migration to the plasma membrane) [6,119,120]; and the DAG-PKC pathway that phosphorylates and thus activates TRPM4 and 5, so ensuring another essential NSCC-mediated shift up to −50 mV [6,111].

### 3.2. Glucagon-Like Peptide 1, GLP-1

#### 3.2.1. GLP-1 Amplification of GSIS

GLP-1 is secreted by enteroendocrine L-cells residing predominantly in the distal ileum and colon upon postprandial stimuli, i.e., by glucose, fatty acids or lipids and proteins [268,269]. GLP-1 results from the expression of preproglucagon (Gcg) and via a specific way of cleaving proglucagon (158 amino acids, AA) in L-cells, different from that secreted by pancreatic α-cells. Thus, proglucagon is cleaved into glicentin (AA 1–69; containing joined glicentin-related pancreatic polypeptide in AA 1–30, glucagon in AA 33–61 and intervening peptide 1, IP1, in AA 62–69) plus GLP-1 (AA 72–107/108), intervening peptide 2 (AA 111–123) and GLP-2 (AA 126–158) [268]. Glicentin can be further cleaved into glicentin-related pancreatic polypeptide and oxyntomodulin (AA 33–69, i.e., joined glucagon and IP1). The most efficient truncated variants are GLP-1_(7–37)_ and its variant GLP-1_(7–36amide)_. The latter is 80% abundant in humans [270]. These variants are termed simply as GLP-1 in this review. Note that the full peptide GLP-1_(1–37)_ is much less efficient at GSIS potentiation [271,272]. In addition, the paracrine GLP-1 signaling acts among the different types of PI cells [271], similarly to the paracrine and endocrine secretion of other hormones. At the systemic level, central control by the brain and nervous system, including GLP-1 secretion in the nucleus tractus solitarii of the brainstem [268], represent the indispensible top level of regulation for insulin secretion, beside local effects of the immune system. GLP-1 effects related to β-cell proliferation or apoptosis are beyond the scope of this review.

Only 10–15% of active GLP-1 likely reaches the pancreas via the circulation system [273]. Thus, typical concentrations of biologically active GLP-1 in human plasma at fasting account for about 2 pmol/L and at most 10 pmol/L postprandially [267], peaking 30–60 min after carbohydrate or protein intake and 120 min after the ingestion of lipids [274].

#### 3.2.2. GLP-1 Receptor Signaling

GLP-1 from the bloodstream acts through its receptor (GLP1R), residing in the plasma membrane of pancreatic β-cells, which is composed of 463 amino acids in humans and rodents [266]. GLP1R activation stimulates Gαs and Gαq/11 and recruits β-arrestin, depending on a biased agonism relatively to different agonists, such as exendin-4 and oxyntomodulin [275,276]. As a scaffold protein, β-arrestin facilitates the signaling via Gαs to cAMP, but also to CREB [276], extracellular regulated kinase ERK1/2 [277] and insulin receptor substrate 2 (IRS-2), its effects promoting β-cell growth, differentiation and maintenance [276]. The stimulation of Gαs leads to the initiation of PKA via enhanced cAMP [278], which also stimulates the EPAC2A pathways [279]. Continuous cAMP production and the partial potentiation of GSIS were found even for the internalized GLP1R [280].

The PKA pathway provides a surplus intracellular Ca^2+^ above that of the net GSIS (i.e., without any receptor stimulation) [281]. Phosphorylation and so deactivation of K_V_ channels prolongs the overall Ca^2+^ stimulation signals. This leads to somewhat lower frequencies of Ca^2+^ oscillations, but with each spike lasting longer [282]. In principle, this phenomenon may also potentiate the 2nd phase of GSIS. In contrast, the PKA-mediated phosphorylation of snapin engages a higher interaction of SNAP-25 with synaptogamins of IGVs, reportedly potentiating the 1st GSIS phase [194,195].

Any phase of insulin secretion within its basic triggering, and its potentiation mechanism by GLP1R signaling, should be dependent on NSCCs. Hence, one can consider that the observed simultaneous stimulation of TRPM2 channels by the EPAC2A pathway [154] is the prerequisite in both. It remains to be resolved whether the TRPM2 stimulation by EPAC activates a greater population of these or other NSCC channels compared to the absence of the GLP1R pathway, i.e., to the basic GSIS. As was mentioned above, the EPAC2A pathway simultaneously promotes the docking and priming of IGVs by allowing Rab3A interaction with Rim2α [205]; and hypothetically also the interaction of EPAC2-Rim2α-Picollo trimers with Rab3A enables more intensive IGV exocytosis [268]. If a biased GLP1R-Gαq/11 stimulation occurs, promotion should also be superimposed by the Ca^2+^-induced (meaning by Ca_V_ opening) RyR-mediated and or IP3R-mediated Ca^2+^ release from the ER [279]; plus activating the phosphorylation of TRPM4 and TRPM5 via the DAG-PKC pathway [219].

However, note that the above description represents a simplification, since it is necessary to understand the detailed contribution of GLP1R signaling to the overall dynamics of Ca_V_ opening (action potential spikes), i.e., to quantify the frequencies, bursting time, and interburst lag time; plus the contribution to [Ca^2+^]_c_ oscillations of superimposed waves originating due to the oscillating Ca^2+^ release from the stores, such as ER, IGVs and/or mitochondria. These events were indeed analyzed by conducting simultaneous electrophysiological and Ca^2+^ oscillation monitoring, the latter using a proper Ca^2+^ fluorescent probe(s) at 2 mM glucose but with 200 μM tolbutamide blocking K_ATP_ [282]. Under these conditions, the GLP-1 analog liraglutide apparently decreased the frequency of action potential spikes, which however were individually wider, as with ablated Kv2.1 channels. So, this most likely reflected the PKA-mediated inactivation of Kv2.1 channels. Independently of liraglutide, each action potential spike corresponded to a triangular peak of the cytosolic Ca^2+^ rise. As a result, the overall integrated [Ca^2+^]_c_ increase over the elapsed time is much higher than upon the net GSIS, which is not amplified by GLP-1.

Earlier experiments with 7.7 mM glucose and using GLP-1_(7–36)_ amide (preproglucagon 78–107) also clearly indicated an increased duration of active and silent electrical activity [283]. Again, the time-width of Ca^2+^ increased from 2 to about 5 s with liraglutide [282]. The relative duration vs. the active phase of Ca^2+^ spikes was 10% at 4 mM, 50% at 7 mM and 80% at 9 mM glucose [284]. This delayed decay of the Ca^2+^ responses is not only given by the prolonged action potential spikes, but also stems from the Ca^2+^ released from the intracellular stores, namely the ER. The intermittent responses of proteins of the exocytotic machinery, namely the frequency of formation of SNARE complexes and resulting pulsatile IGV exocytosis, was elegantly monitored by the ATP-activated currents conducted by the overexpressed P2X2 cation channels [285]. This was made possible by the fact that IGVs contain a high ATP concentration.

#### 3.2.3. Two Phases of Insulin Secretion vs. GLP-1 Potentiation

Both cAMP and 8-Br-cAMP increased the frequency of fusion events, i.e., the fusion of insulin granules to the plasma membrane in both phases of GSIS, in experiments using TIRF while simulating GLP-1 effects [85]. EPAC2A interacts with the small G protein Rap1, affecting its conformation so that it releases the catalytic region, which subsequently binds and thus activates another G protein Rap113. In EPAC2A knockout mice, most of the potentiation of the 1st GSIS phase vanished [85]. Thus, one can speculate that the 2nd phase amplification can be predominantly upregulated due to the PKA pathway.

#### 3.2.4. Does GLP-1 Stimulate Insulin Secretion at Low Glucose?

The traditional view has considered that incretin signaling does not stimulate insulin release under low glucose conditions [271,272]. However, in experiments mimicking the physiological situation “low glucose” should be equal to fasting glycemia. Under these conditions, however, GLP-1 in supraphysiological concentrations did initiate insulin secretion. The controversies are therefore over whether reported experiments reflect the physiologically occurring postprandial GLP-1_(7–37)_ concentration in islet capillaries of rodents (GLP-1_(7–36amide)_ in humans).

It has been observed that the intermediate glucose concentration, such as 6 mM, allows certain insulin secretion in rats as the response to GLP-1_(7–37)_, but, in contrast, at < 2.6 mM glucose GLP-1_(7–37)_ does not stimulate insulin secretion [271,272]. Note that 6 mM glucose approaches to the fasting glycemia values in rats. As described above, the store-release of Ca^2+^ should depend on the Ca_V_ opening, but this comes automatically, since both GLP1R signaling branches, PKA and EPAC2 pathways, prolong durations of Ca_V_ opening. PKA does this directly by phosphorylating Ca_V_ (besides K_V_ and K_ATP_) and EPAC2 provides the basic or hypothetically even the amplified TRPM2 action. Theoretically, the GLP1R signaling can be considered to be independent of the high glucose concentrations, since the basic shift to −50 mV is provided by the EPAC2-activated TRPM2 [154] and since the PKA phosphorylation of K_ATP_, Kv and especially of Ca_V_ channels might provide the triggering. If all these considerations turned out to be plausible, we must ask the question of whether all the proteins of IGV exocytotic machinery could function at the Ca^2+^ levels established at these low or intermediate glucose concentrations. Indeed, certain synaptogamins were reported to be independent of Ca^2+^ [230].

### 3.3. Glucagon Inhibitory Peptide, GIP

GIP acts similarly to GLP-1 [286]. GIP is secreted by the K-cells in the proximal gut upon a stimulus when chyme enters the duodenum. Similarly to GLP-1, it has been reported that the GIP-amplification of GSIS is dependent on higher glucose levels, cAMP production due to the GIP receptor (GIPR) signaling is not [287,288]. The pathways downstream of GIPR are thought to be identical to those for GLP-1. For example, Rim2α knockout mice exhibited impaired GIP amplification of GSIS, indicating the absence of the GIPR-cAMP-EPAC2-enhancement of IGV exocytosis [205]. Similarly to GLPR, GIPR long-term signaling exhibits profound prosurvival effects for pancreatic β-cells [289].

## 4. Role of Redox Shuttles upon GSIS

### 4.1. Redox Shuttles Exporting Redox Equivalents to the Cytosol of Pancreatic β-Cells

Even without knowledge of NOX4 participation in GSIS, it has long been known that an increase in cytosolic NADPH facilitates insulin secretion responding to glucose [16,37,290,291,292,293]. The three major metabolic shuttles were revealed, which are activated at higher glucose and export reducing equivalents of NADH from the mitochondrial matrix to the cytosol during GSIS [290]. Since the major metabolite influx into mitochondria upon increasing glucose is given by pyruvate, all three shuttles are related to pyruvate [294]. One can recognize the pyruvate/malate, pyruvate/citrate and pyruvate/isocitrate shuttles (Figure 6).

#### 4.1.1. Pyruvate/Malate Redox Shuttle

The reaction sequence for the pyruvate/malate redox shuttle bypasses the regular pyruvate entry into the Krebs cycle via pyruvate dehydrogenase (PDH). Instead, the pyruvate carboxylase reaction takes place. The resulting oxaloacetate can either increase the turnover of the forward Krebs cycle or may be converted by the reverse reaction of malate dehydrogenase (MDH) to malate. This counter-Krebs cycle flux is the key element in the pyruvate/malate shuttle. Malate can be subsequently exported to the cytosol, either by the 2-oxoglutarate carrier (SLC25A11) [292], being exchanged for 2OG import into the mitochondrial matrix. The exported malate aids the cytosolic malate pool. This pool can be consumed by the malic enzyme 1 (ME1) reaction, driven by NADP^+^, so being one of the cytosolic sources of NAPDH [295,296]. The proper reaction direction within the β-cell cytosol is driven by an immediate consumption of pyruvate by mitochondria. So, the pyruvate influx into the matrix, enabled by the pyruvate carrier (MPC1 and MPC2; providing pyruvate-H^+^ symport) both begins and ends the pyruvate/malate shuttle.

#### 4.1.2. Pyruvate/Citrate Redox Shuttle

The pyruvate/citrate shuttle exists due to citrate export from the matrix, while the Krebs cycle is truncated just after the reaction of citrate synthase [297]. ^13^C-tracing studies showed that the predominant amount of citrate originates from glucose-derived acetyl-CoA in INS/1 832/13 cells, indicating that a quite high fraction of citrate escapes the Krebs cycle [62]. Such an export exists frequently in cancer cells. However, it is also of great importance for GSIS. Citrate is exported by the citrate carrier (SLC25A1), enabling citrate antiport with malate. Note that the malate cycling can actually take place in conjunction with the pyruvate/malate redox shuttle. In the cytosol, citrate is split by the ATP-citrate lyase (ACL) in the reaction with coenzyme A (CoA) into oxaloacetate and acetyl-CoA (AcCoA). Cytosolic MDH converts oxaloacetate into malate and also this shuttle uses ME1 and a pyruvate carrier to produce and import pyruvate into the matrix, respectively. In parallel, ME1 supplies NADPH for NOX4-mediated redox signaling. Note that the ACL reaction may be substituted by the acetoacetate pathway (see Section 4.2.5), so that at low glucose either one maintains the short-chain acyl-CoA levels in the cytosol, but at GSIS the acetoacetate pathway predominates [298].

#### 4.1.3. Pyruvate/Isocitrate Redox Shuttle

The pyruvate/isocitrate shuttle is based on the reductive carboxylation reaction of isocitrate dehydrogenase 2 (IDH2), residing in the mitochondrial matrix. This is a reverse reaction compared to the typical NADP^+^-driven isocitrate oxidation, and produces 2OG and NADPH. This oxidative IDH2 reaction serves as a parallel bypass of the regular IDH3 reaction of the Krebs cycle. However, instead of NADH produced by IDH3, IDH2 forms NADPH.

Upon a higher glucose intake, the forward (oxidative) mode is reversed to the counter-Krebs cycle direction. This reductive carboxylation mode requires CO_2_ and enables the operation of the pyruvate/citrate cycle [37]. At the same time, the aconitase reaction and isocitrate formation is slow, thus allowing the reverse IDH2 reaction, despite the CO_2_ requirement. The citrate carrier then readily exports isocitrate to the cytosol. Subsequently, the cytosolic NADP^+^-driven IDH1-mediated oxidation of isocitrate takes place, so 2OG and NADPH are produced [295]. IDH1 within this shuttle is an important source of the cytosolic NADPH pool and contributes to its increase upon GSIS [293]. 2OG enters the matrix by being exchanged for malate via by the oxoglutarate carrier. 2OG then either enters the regular Krebs cycle 2OG-dehydrogenase reaction; or 2OG completes this cycle, again being the substrate of IDH2-mediated reductive carboxylation.

### 4.2. Specific Metabolism at Insulin-Non-Stimulating vs. Insulin-Stimulating Glucose in Pancreatic β-Cells

#### 4.2.1. NADPH and NADH Homeostasis in Pancreatic β-Cells

The above detailed description of redox shuttles considered their functioning at high, insulin-stimulating glucose, i.e., at maximum metabolic turnover and maximum OXPHOS. As a result, these shuttles do not allow maximum NADH to be produced in the mitochondrial matrix, but instead, more NADPH is produced in the cell cytosol. In fact, this represents a transfer of redox equivalents from the matrix to the cell cytosol. The shuttles thus serve as an independent NADPH source feeding NOX4 to initiate redox signaling that enables K_ATP_ closing (and/or TRPM2 opening), resulting in insulin secretion.

For example, as part of the pyruvate/malate redox shuttle, mitochondrial MDH produces less NADH, than with the 100% forward reaction. However instead, malate produced by the reverse MDH reaction is used by ME1 to form NADPH in the cytosol. Similarly, if a portion of the isocitrate is not converted by IDH3 to produce NADH, but instead the reverse (reductive carbonylation) IDH2 reaction occurs, while isocitrate exported into the cytosol evokes NADPH formation by IDH1. Again, instead of the production of one NADH molecule in the matrix, one NADPH molecule is formed in the cytosol. In order to facilitate the exocytosis of IGVs by glutamate, the reported glutamate dehydrogenase mode in the matrix should involve the reaction producing glutamate and NAD^+^, while consuming 2OG, ammonium and NADH [66]. One can predict that this reaction will also contribute to the decreasing NADH in the matrix.

We predicted and verified experimentally [37] that the decreasing matrix NADH during the concomitantly increasing NAD^+^ at the simultaneous maximum respiration leads to a decreased matrix substrate pressure *SP*, defined as the NADH/NAD^+^ ratio in the mitochondrial matrix. Using inhibitors of superoxide formation at specific RC sites, we have provided evidence that even slightly decreasing *SP* reduces superoxide formation at the flavin site I_F_ of RC complex I [37]. Note, however, that a typical matrix concentration of NAD^+^ was estimated to be two orders of magnitude higher than the NADH concentration. In HeLa cells, for example, 800 μM NAD^+^ was found besides 5 μM NADH [299]. For pancreatic β-cells, set to a faster respiration upon GSIS, this means that each NADH molecule formed by the respective matrix dehydrogenases is immediately consumed and used by complex I.

#### 4.2.2. Nicotine Nucleotide Translocase in Pancreatic β-Cells

Somewhat substantial concerns were raised regarding the mitochondrial nicotine nucleotide transhydrogenase (NNT) in pancreatic β-cells. In its forward mode, NNT consumes Δp by allowing H^+^ influx into the matrix, which is coupled with the conversion of NADP^+^ to NADPH with simultaneous NADH consumption, thus producing NAD^+^ [300]. Note that NNT acts downstream of the redox shuttles, so it should not influence them. However at low glucose, NNT might act in the reverse mode [301], in which it functions as a proton pump, enhancing Δp and consuming NADPH and NAD^+^, while producing NADP^+^ and NADH. At low glucose, the respiration and ATP synthesis are at slower rates; establishing a lower Δp. Therefore, H^+^ pumping mediated by NNT against an intermediate Δp would be hypothetically easier than at high glucose against the higher Δp. The existence of the NNT reverse mode was suggested, based on different types of experiments with mouse pancreatic β-cells [301]. To our knowledge, the direct observation of the H^+^ flux direction by NNT has never been attempted. Our preliminary data concerning the comparison of experimentally determined Δ*Ψ*_m_, using fluorescent indicators of membrane potential, showed instead increases in Δ*Ψ*_m_ in NNT-silenced INS-1E cells [37]. This suggests that NNT acts in the forward mode, producing NADPH.

Experiments relying on a comparison of C57BL/6J vs. C57BL6/N mice were also controversial. In C57BL/6J mice, an in-frame five-exon deletion in the *Nnt* gene spontaneously occurred, removing exons 7–11 and leading to a complete absence of the NNT protein [302,303,304]. It was reported that there is a high blockage of GSIS in the C57BL6/J mouse strain, which others and we never observed [4,305]. The discrepancy may originate from the fact that initially only the quantitative trait loci (QTLs) were identified and correlated with deletions in the *Nnt* gene. The subsequent verification using artificial *Nnt* expression could be inconclusive, since the *Nnt* expression per se could enhance insulin secretion, leading to its apparent suppression in C57BL6/J mice [305].

#### 4.2.3. Mitochondrial Matrix NADPH Homeostasis in Pancreatic β-Cells

It has also been observed that matrix NADPH decreases with operating redox shuttles, specifically due to the operation of the pyruvate/isocitrate shuttle. Such an NADPH decrease was confirmed experimentally in INS-1E cells [37]. Hence, even if the NADPH formation was increased upon GSIS by the activated matrix malic enzyme ME3 and by the increasing forward (Δp consuming) mode of NNT, the latter two sources cannot exceed the NADPH consumption by the mitochondrial IDH2. ME3 was reported to form pyruvate and NADPH from malate and NADP^+^ in the mitochondrial matrix [306]. The acute NADPH increase has a disadvantage in that it decreases the reduced glutathione (GSH) and increases its oxidized form (GSSH) in the mitochondrial matrix. One can view the simultaneous existence of redox shuttles with decreasing GSSH as a fair trade for GSIS, lying in the preferable provision of cytosolic NADPH elevations prior to the maintenance (or at the expense of) mitochondrial matrix GSH.

#### 4.2.4. Relationships to the Malate/Aspartate Shuttle

A major role has been suggested for the malate/aspartate shuttle in pancreatic β-cells [279,307]. However, one must recognize that the active malate/aspartate shuttle operation upon GSIS excludes the operation of the pyruvate/malate and pyruvate/isocitrate shuttles [37], the existence of which was documented by numerous experiments [16,275,276,277,278]. The malate/aspartate shuttle could not effectively function under high glucose conditions, since the 2-oxoglutarate carrier would have to translocate malate in the opposite direction to the malate export, which is active during the pyruvate/malate and pyruvate/isocitrate redox shuttles. Additionally, as part of the malate/aspartate shuttle, glutamate must be imported to the mitochondrial matrix, coupled to the aspartate efflux. However, several laboratories reported that matrix-formed glutamate is also exported to the β-cell cytosol to facilitate IGV maturation and exocytosis [306,308,309,310]. This again requires the opposite direction of fluxes. At least glutamate is beneficial for the GLP-1 amplification of GSIS [67,311].

We found that if there is any contribution of the malate/aspartate shuttle, it must act at low non-stimulating glucose [37]. Previously, the support for the existence of the malate/aspartate shuttle was considered to stem from the essential requirement of transaminases (aminotransferases) and two isoforms of aspartate/glutamate antiport carrier: SLC25A12, i.e., Aralar or AGC1 [312,313]; and SLC25A13, i.e., AGC2. Metabolomics studies showed that the total cell aspartate is diminished at the initiation of GSIS, while aconitate, citrate, isocitrate, malate or fumarate rose instantly, and elevations of 2-OG or succinate were delayed by 15 min [38]. However, elevations of metabolites can originate from the delayed reaction consuming that particular metabolite, such as acting in the malate/aspartate shuttle.

#### 4.2.5. Acetoacetate and β-Hydroxybutyrate in Pancreatic β-Cells

The β-hydroxybutyrate dehydrogenase (β-OHBDH), which is found exclusively in the mitochondrial matrix, was also studied in pancreatic β-cells [116,314]. The previous consensus for hepatocytes considered that the ratio of β-hydroxybutyrate to acetoacetate reflects the established matrix NAD^+^/NADH ratio (1/*SP*) [314]. However, since the estimated order of magnitude for 1/*SP* is >100 [299], such a ratio of β-hydroxybutyrate to acetoacetate is unlikely. Additionally, the NADH-driven β-OHBDH reaction converting acetoacetate to β-hydroxybutyrate and NAD^+^ is not the only one involving acetoacetate in pancreatic β-cells [116,314]. Acetoacetate is also exported to the cytosol of β-cells, where it reportedly facilitates insulin secretion via forming various acyl-CoA derivatives [116]. Additionally, β-OHBDH cannot be formed in the cytosol of rodents, but can be formed in the β-cell cytosol in humans (for further details see https://www.brenda-enzymes.org/enzyme.php?ecno=1.1.1.30). All these reactions influence redox homeostasis.

Under high glucose conditions, the sufficient succinate load enables its interconversion by succinyl-CoA:3-ketoacid-CoA transferase (SCoA:3oxoAcCoAT) with acetoacetyl-CoA (AcAcCoA) into succinyl-CoA (Succ-CoA) and acetoacetate [298]. This can also be formed by leucine metabolism, within a series of reaction of β-like oxidation leading to hydroxymethyl-glutaryl-CoA (HMG-CoA), which is split by HMG-CoA lyase (HMG-CoAL). From the acetoacetate pool in the matrix, acetoacetate can be easily transported to the cytosol or may be converted by β-OHBDH to β-hydroxybutyrate at the expense of NADH, thus forming NAD^+^. Note that distinct enzyme isoforms convert two molecules of acetyl-CoA (Ac-CoA) into CoA and AcAcCoA in the mitochondrial matrix (acetyl-CoA acetyltransferase, ACAT1; and acetyl-CoA acyltransferase, ACAA2) and cytosol (acetyl-CoA acetyltransferase, ACAT2; and acetyl-CoA acyltransferase, ACAA1). Cytosolic acetyl-CoA was suggested to facilitate the acetylation of proteins, which might speculatively enhance GSIS [304,305]. In conclusion, acetoacetate and acyl-CoA metabolism should be studied further, not only revealing metabolomics data, but also in order to elucidate in detail whether they amplify/stimulate insulin secretion or not.

#### 4.2.6. Phosphoenolpyruvate Shuttle and the Role of Pyruvate Kinases

Yet another cycle may be important for pancreatic β-cells in the resting regime, when no insulin secretion takes place. This is the so-called phosphoenolpyruvate (PEP) cycle, which is cataplerotic in character. It begins when the mitochondrial matrix oxaloacetate is converted to PEP by the mitochondrial PEP-carboxykinase (PEPCK2). PEP is subsequently exported to the cytosol by the citrate carrier (SLC25A1). There are several cytosolic isoforms of pyruvate kinase (PK) in β-cells, such as the constituent M1 isoform and recruitable M2 and L [315]. PKs convert cytosolic PEP to pyruvate. This reaction is coupled to ATP formation by the “substrate” phosphorylation of ADP and is the regular part of glycolysis. That is why at high glucose, glucose metabolism is highly competitive and the PEP cycle should contribute to a lesser extent. Nevertheless, under both low and high glucose conditions, pyruvate then enters mitochondria, where it is metabolized either by PDH or PC. The PC flux completes the PEP cycle by the conversion of pyruvate to oxaloacetate.

Since fructose 1,6-bisphosphate allosterically activates PKM2 and PKL, their reactions supply the ATP required for K_ATP_ closure. This was recently derived from the results of patch-clamp experiments in the excision mode combined with PK activation by a small molecule activator [100]. The PEP-cycle was reportedly switched on/off in the β-cell responses to intermediate 9 mM glucose, when *V*p and/or [Ca^2+^]_c_ bursts phases are intermittent with the interburst phases (cf. Figure 3). The decreased cytosolic ATP/ADP-ratio was explained on the basis of the PEP cycle providing ATP by PKs, which should be independent of OXPHOS. Prior to glucose elevation, or after termination of the burst phase at 9 mM glucose, lower ATP levels were observed, which set K_ATP_-channels open. Under these conditions, the control strength of the PEP cycle is high. When OXPHOS continues at this intermediate 9 mM glucose concentration, thus continuing the elevation of ATP, the control strength of the Krebs cycle overcomes that of the PEP cycle and it possesses less control strength. As a result, at this moment the burst phase begins at 9 mM glucose. Additionally, a PKM2 and PKL activator failed to improve GSIS in PEPCK2-knockout mice, demonstrating the important contribution of the PEP cycle [316].

## 5. Mechanism of Insulin Secretion Stimulated by Branched-Chain Ketoacids

A postprandial response of insulin secretion is also produced by substances other than glucose, which induce the secretion of insulin. These substances are termed secretagogues in general. One important type of secretagogue is branched-chain ketoacids (BCKAs). These are metabolites of branched-chain amino acids (BCAAs). BCKAs involve three major metabolites, 2-ketoisocaproate (KIC; leucine metabolite) [317], 2-ketoisovalerate (KIV; valine metabolite) and 2-ketoisomethylvalerate (KMV; isoleucine metabolite) [318,319,320,321,322]. We will discuss mechanisms leading to redox signaling upon BCKA metabolism [323].

### 5.1. Superoxide Formation and Subsequent Redox Signaling upon β-Like Oxidation of Branched-Chain Ketoacids

#### 5.1.1. Electron Transfer Flavoprotein: Ubiquinone Oxidoreductase (ETFQOR) as the Key Factor

The metabolism of BCKAs is initiated in the mitochondrial matrix by the BCKA dehydrogenase complex (BCKDH; not present in the cytosol), following a similar sequence of reactions to the β-oxidation of fatty acids, termed β-like oxidation. The BCKDH reaction [323] within the β-like oxidation leads to the elevated formation of superoxide in the mitochondrial matrix. Indeed, the silencing of the BCKDH E1α subunit prevented both the KIC-induced matrix superoxide release and insulin secretion stimulated with KIC in INS-1E cells [4].

The BCKDH complexes provide reactions of isovaleryldehydrogenase (IVD, EC 1.3.99.10) and similar dehydrogenases of isobutyryl-CoA and methyl-isobutyryl-CoA, all having FAD as a cofactor, converting it to FADH_2_. Subsequently, the electron-transfer flavoprotein (ETF) accepts electrons from BCKDH as one electron carrier and diffuses up to the electron-transfer flavoprotein: ubiquinone oxidoreductase (ETFQOR) located in the matrix surface of IMM [324,325].

Superoxide formation may originate due to the transfer of electrons from the ETFQOR to the mitochondrial ubiquinone pool [325]. Either an electron leak from flavin to the oxygen leading to radical pair formation produces superoxide within ETFQOR itself [326]; or since ETFQOR needs ubiquinon (Q), reduced after accepting two electrons from the ETF to ubiquinol (QH_2_), it effectively competes with the mitochondrial respiratory chain complex I. Since QH_2_ binds to the complex I Q-binding site, it allows a feedback inhibition of the ongoing Q reduction to QH_2_ within complex I. This increases the mitochondrial superoxide formation at the I_Q_ site of superoxide formation [312]. In other words, the electron transfer within complex I is effectively retarded in this way and therefore superoxide formation takes place in its I_Q_ site. Brand and colleagues suggested that dehydrogenases alone can also produce superoxide [327]. However, this is yet to be confirmed or excluded for BCKDH. Additionally, superoxide may be formed by complex I and complex III of the mitochondrial respiratory chain due to the resulting excessive acetyl-CoA entry (propionyl-CoA entry for KIV; through methylmalonyl and succinyl-CoA) into the Krebs cycle. Moreover acetoacetate, being one of the final products of leucine metabolism, could influence the established redox homeostasis.

#### 5.1.2. Redox Signaling upon β-Like Oxidation of Branched-Chain Ketoacids

Independently of the molecular mechanism, BCKA metabolism leads to increased mitochondrial superoxide formation. After conversion to H_2_O_2_ by the matrix MnSOD and the intermembrane space CuZnSOD, the ongoing H_2_O_2_ efflux from mitochondria can be regarded as redox signaling. We have clearly demonstrated that the absence of such redox signaling, otherwise stimulated with BCKAs, leads to a blockage of insulin secretion, for example in the presence of the mitochondrial matrix-targeted antioxidants SkQ1 [4]. Likewise, the silencing of BCKDH led to the inhibition of insulin secretion stimulated with BCKAs.

In summary, superoxide formed in an enhanced manner upon β-like oxidation is converted to elevated H_2_O_2_. The emanation of H_2_O_2_ from the matrix to targets, located e.g., in the plasma membrane, subsequently substantiates the mitochondrial retrograde redox signaling. The targets providing insulin secretion could be K_ATP_, TRPM2 or both, as with GSIS. Simultaneously, the metabolism of BCKA [328,329], followed by OXPHOS, leads to the concomitantly elevated ATP (Figure 7). Note also that the redox signaling can take place either by direct diffusion or hypothetically also via redox relay systems (see Section 7).

### 5.2. Metabolism of Branched-Chain Ketoacids

#### 5.2.1. Branched-Chain Amino Acids as Precursors of Branched-Chain Ketoacids

Leucine, valine and isoleucine are imported into β-cells by various transporters, predominantly by LAT1/SLC7A5 [330]. BCAA metabolism proceeds in mitochondria, since there is no branched-chain amino acid aminotransferase (BCAT) in the β-cell cytosol [328,329] (Figure 7). Additionally, the nature of BCAA import into the mitochondrial matrix is unknown. BCKAs are most likely imported into the mitochondria of pancreatic β-cells by the carnitine carrier (SLC25A20). The mitochondrial matrix BCAT with a cofactor (pyridoxal-5′-phosphate) reversibly converts BCAAs and 2OG to BCKA and glutamate [328,329]. The resulting BCKAs are degraded by the mitochondrial branched-chain ketoacid dehydrogenase (BCKDH) complex [323,331]. This reaction is coupled to the conversion of CoA to proper BCKA-CoAs.

#### 5.2.2. β-Like Oxidation of Branched-Chain Ketoacids

BCKDH forms isovaleryl-CoA, isobutyryl-CoA and methyl-isobutyryl-CoA from KIC, KIV and KMV, respectively. The next series of reactions is similar to the mitochondrial β-oxidation of fatty acids, hence is termed β-like oxidation. The sequence of reactions for KIC is given by the enzymes in the following order: isovaleryl-CoA dehydrogenase (IVD, EC 1.3.99.10), methylcrotonyl-CoA carboxylase (MCC), methyl-glutoconyl-CoA hydratase (MGCoAH) and 3-hydroxy- 3-methylglutaryl-CoA lyase (HMGCoAL). The end products are acetyl-CoA and acetoacetate. Similarly as for pyruvate metabolism via PDH, the common end product acetyl-CoA drives the Krebs cycle. This metabolic acceleration itself may increase mitochondrial superoxide formation.

The BCKDH complex consists of three catalytic components [332]: α-ketoacid dehydrogenase (E1), dihydrolipoyl transacylase (E2) and dihydrolipoamide dehydrogenase (E3). In total 24 copies of E2 are arranged in octahedral symmetry and form the core [333]. Twelve E1 α2β2 tetramers are non-covalently linked to the core as well as six E3 homodimers. The inner-core domain catalyzes the acyltransferase reaction. The BCKDH complex is also indirectly activated by Ca^2+^ via a mechanism that cancels the inhibitory phosphorylation of the E1 subunit. E1P-phosphatase dephosphorylates the E1 subunit, while phosphorylation is enabled by the BCKDH-E1-kinase that is in turn inhibited by the cofactor thiaminepyrophosphate. The cofactor-mediated kinase inhibition is strengthened by Ca^2+^, hence Ca^2+^ activates BCKDH [334].

### 5.3. Insulin Secretion Stimulated by Branched-Chain Ketoacids

#### 5.3.1. Overview of Branched-Chain-Ketoacid-Stimulated Insulin Secretion

One can envisage that the rules for GSIS are also valid for BCKA-stimulated insulin secretion. In both cases, the increased metabolism and Krebs cycle turnover resulting in elevated ATP synthesis is paralleled by the redox signaling. This stems from mitochondria for BCKA-stimulated insulin secretion or originates from the NOX4 reaction for GSIS. When both conditions are fulfilled, i.e., ATP plus H_2_O_2_ is elevated, this leads to conditions required for the Ca_V_ opening and IGV exocytosis (either ATP plus H_2_O_2_ are required for the K_ATP_ closure; or ATP closes K_ATP_ plus H_2_O_2_ activates some of NSCCs, such as TRPM2).

Additionally, the Ca^2+^ activation of BCKDH has an important self-accelerating role when the BCKA-stimulation of insulin secretion is initiated. The primary elevations of the cytosolic [Ca^2+^]_c_ due to the Ca_V_ opening are relayed to the increases in mitochondrial matrix [Ca^2+^]_m_ [99] and consequently activate BCKDH. So, in the next “turn”, more intensive OXPHOS and redox signaling lead to more intensive insulin secretion.

#### 5.3.2. Physiological Context of Branched-Chain-Ketoacid-Stimulated Insulin Secretion

BCAAs were previously found to stimulate insulin secretion in vivo in numerous reports, without a consensus on the mechanism. However, it was recognized that BCAAs can either amplify GSIS or act as secretagogues per se. The latter was confirmed to be based on the oxidative decarboxylation of the respective BCKAs [335]. Additionally, the ability of leucine to allosterically activate glutamate dehydrogenase was considered [335,336].

The physiological context and relationships between the BCKA-stimulated insulin secretion and GSIS should be the subject of further studies. One can predict GSIS to be faster and BCKA-stimulated insulin secretion to be delayed from the onset of maximum insulin theoretically given by the net GSIS after a meal rich in saccharides and proteins. In contrast, the most typical situation for BCKA-stimulated insulin secretion can occur during fasting, when there are maximum levels of BCKAs in blood plasma.

## 6. Mechanism of Fatty Acid-Stimulated Insulin Secretion

### 6.1. Relevancy of Fatty Acid-Stimulated Insulin Secretion (FASIS)

#### 6.1.1. Experimental Approach vs. Physiology

It is still controversial, as to whether there is fatty acid stimulated insulin secretion (FASIS) at low (insulin non-stimulating) glucose concentrations. Some previous observations suggested that glucose should always be present for FAs to induce any insulin secretion response [116,313,314,337]. However, several other observations described FASIS at low glucose concentrations (which alone do not stimulate insulin secretion) [4,27,338,339], but being absent at zero glucose [339]. Nevertheless, human islets perfused at zero glucose do not increase their respiration upon the addition of long-chain FAs, but do release insulin [339]. Moreover, both respiration and insulin release were increased when long-chain FAs were added to human islets in the presence of 5.5 mM glucose. These results fit well with FASIS having two components: a metabolic one and a component dependent on metabotropic receptor signaling [338].

Of course, one should consider the above-mentioned testing conditions with zero glucose as completely non-physiological. Since FAs trigger the action potential at a low, insulin non-stimulating glucose concentration in pancreatic β-cells [340], FASIS should be studied independently of GSIS. Physiologically, postprandial responses should exist due to all secretagogues resulting from major saccharide, fat and protein components. A mixed or fatty meal leads to the intestinal formation of chylomicrons. However, chylomicrons are brought by the circulation system to pancreatic islets with a delay, which amounts to a few hours in humans [341] and at least a half an hour to an hour in rodents [342,343]. Some authors reported that the lipid- or FA-mediated secretion of GLP-1 by intestinal L-enterocytes is instead instantaneous [344]. This contrasts with the findings of peaks in circulation for biologically active GLP-1 in human plasma 30–60 min after carbohydrate or protein intake and 120 min after the ingestion of lipids [274]. Hence it needs to be verified for humans whether the peak of GLP-1 coincides with the peak of chylomicrons in circulation, or whether GLP-1 precedes the peak of chylomicrons. The same is needed for rodents. This is important, because knowledge of precise timing can distinguish between GLP-1-amplified GSIS at intermediate glucose from the net FASIS. At least FAs mobilized from white adipose tissue stores should not coincide with the above phenomena, since lipolysis is inhibited by the secreted insulin [341].

Nevertheless, it seems plausible that a delayed insulin secretion due to chylomicrons should be expected when even the one-hour-lasting 2nd GSIS phase is nearly complete in rodents. At this time, the inhibition of lipolysis ceases due to the return of blood insulin levels and glycemia back to “fasting” values before the meal. However, the incoming chylomicrons should stimulate at least some response, and these responses would thus occur already at “fasting” glucose concentrations. The GLP-1 amplifying component (initiated due to intestinal lipids or FAs) may precede FASIS, especially if it still occurs during intermediate glycemia. Whether simultaneous responses to GLP-1 and FAs occur in vivo should also be studied. In conclusion, not only due to molecular mechanistic reasons, but due to the physiological timing, it is crucial to study FASIS experimentally. Of course, in supraphysiological experiments FASIS may overwhelm responses to the other secretagogues, especially when applied intraperitoneally or intravenously or when originating from an experimental high-fat diet [345].

#### 6.1.2. GPR119 Pathway in Pancreatic β-Cells

Physiologically, one cannot separate FASIS from the concomitant portion of insulin secretion stimulated via 2-monoacylglycerols (MAGs). MAGs are agonists for another metabotropic receptor, GPR119, which provides signaling via Gαs and cAMP [184,263]. Thus, all pathways relevant, e.g., for GLP1R are activated by GPR119. Additionally, the metabolic component of FASIS undoubtedly involves FA β-oxidation (see below), providing both ATP from the elevated OXPHOS and H_2_O_2_ from the enhanced superoxide formation by the respiratory chain and due to the ETFQOR input of QH_2_ to it, as with BCKAs. In conclusion, MAG-stimulated insulin secretion via GPR119 is also allowed due to the metabolic component, which can be provided by FAs under conditions of fasting glucose. Nevertheless, MAG-stimulation representing an amplification of GSIS is the end point of the so-called glycerol/FA cycle, described below.

#### 6.1.3. Physiological Stimulation of GPR40 and GPR119 Receptors in Pancreatic β-Cells

Fatty acids appear in pancreatic islet capillaries either bound to albumin, being components of lipoproteins of endogenous sources, or part of postprandial chylomicrons resulting from dietary fat lipids. The latter are rich in triglycerides, which are cleaved locally in pancreatic islet capillaries by lipoprotein lipase [318]. The resulting MAGs and long-chain FAs [338,346,347,348] each stimulate their own receptor [270]. Adipose triglyceride lipase (ATGL) was reported to be the major isoform cleaving these triglycerides [349].

Secretory phospholipases A2 might also contribute to the free pool of long-chain FAs in the islet capillaries, in addition to PLC, residing on the plasma membrane and being inherent in GPR40 signaling [350]. In parallel, long-chain FAs are imported into pancreatic β-cells by the CD36 FA transporter. A fraction of CD36 molecules may be acetylated at several lysines, which blocks its function. Therefore, at elevated NAD^+^, which only occurs upon GSIS, sirtuins are activated and deacetylate the particular CD36 fraction [351]. As a result, the import of free long-chain FAs is promoted. Even this mechanism can be regarded as a certain type of potentiation of FASIS upon GSIS.

#### 6.1.4. Physiological Stimulation of Other Metabotropic Receptors in Pancreatic β-Cells

Short-chain fatty acids act specifically via the GPR41 metabotropic receptor and contribute to a fine tuning of insulin secretion in both fed and fasting states [352,353]. Similarly, the metabotropic receptor GPR120, which has a different selectivity for agonists, mediates the amplification and/or stimulation of insulin secretion, by, e.g., α-linolenic acid and polyunsaturated FAs [354].

### 6.2. GPR40 and Metabolic Pathway upon Fatty Acid-Stimulated Insulin Secretion

#### 6.2.1. GPR40 Pathway in Pancreatic β-Cells

Theoretically, FASIS may contain one component that depends on the metabolism and a second component that relies on stimulation by the metabotropic receptor GPR40 [27,184,338,355,356,357,358,359,360,361,362]. Its activation is not involved in the FA chronic induction of lipotoxicity [363]. The FASIS mechanism that is dependent on GPR40 activation should be a major one, since the ablation of GPR40 or its point R258W mutation impaired FASIS [358].

The major pathway downstream of GPR40 acts via Gαq/11, activating PLC, releasing IP3 and DAG. Therefore, the most prominent pathway for FASIS under low glucose conditions should be GPR40-Gαq/11-PLC-DAG-PKC, phosphorylating TRPM4 (TRPM5) channels and activating them, which would aid the necessary shift to the depolarization by the 100% closed K_ATP_ ensemble. The K_ATP_ closure should be ensured by the metabolic component of FASIS, which might, as illustrated in Figure 1 and Figure 8, either cause closing of the entire K_ATP_ ensemble; and simultaneously or alternatively to this, could activate TRPM2. The GPR40-Gαq/11-PLC-DAG pathway, signaling via PKC, reportedly leads to the phosphorylation and activation of another NSCC channel, TRPC3 [359]. The routes involving GPR40-Gαq/11-PLC-IP3 activate the additional Ca^2+^ efflux from the ER initiated by the Ca_V_ opening [345].

The GPR40 receptor signaling also activates pathways of the signal-regulated kinase 1 and 2 (ERK1/2) [360]. Moreover, the downstream signaling of the GPR40 receptor slightly increases respiration [361]. GPR40 signaling was also implicated in the activation of p21-activated kinase 4 (PAK4) [364]. Since PAK4 was known to be a regulator of cytoskeletal dynamics, this pathway may aid the facilitation of IGV exocytosis. Additionally, protein kinase D (PKD) was confirmed to be activated downstream of GPR40 and DAG [365].

#### 6.2.2. β-Oxidation of Fatty Acids in Pancreatic β-Cells

During GSIS, and because of the operation of the pyruvate/isocitrate and pyruvate/citrate shuttles, conditions are set to promote FA synthesis. Accordingly, metabolomics studies indicated a palmitic acid increase after the transition from low to high glucose conditions [307]. Note also that acyl-CoAs, as intermediates of FA β-oxidation, were suggested to stimulate/amplify insulin secretion [7]. In contrast, under low glucose conditions, FA β-oxidation readily proceeds in pancreatic β-cells, and therefore supplies the cell’s needs when cells are depleted of glucose. Acyl-CoA synthetase converts FAs to acyl-CoAs, while the cytosolic carnitine acyltransferase 1 (CAT1) converts acyl-CoAs to acyl-carnitines [366]. Their uptake into the mitochondrial matrix is provided by the carnitine carrier (SLC25A20), which imports long-chain acyl carnitines in exchange for carnitine. Next, the matrix carnitine acyltransferase 2 (CAT 2) provides the conversion of acyl carnitines to acyl-CoAs. The following chain of reactions shortens the FA acyl chain by two carbons, involving acyl-CoA dehydrogenases, enoyl-CoA hydratase, 3-hydroxyacyl-CoA dehydrogenase and β-thiolase. The product of a single cycle is just acyl-CoA shortened by two carbons plus acetyl-CoA.

#### 6.2.3. β-Oxidation of Fatty Acids Produces Superoxide

In general, FA β-oxidation involves mitochondrial flavoprotein dehydrogenases, such as the short-chain acyl-CoA dehydrogenase (EC 1.3.99.2); medium-chain acyl-CoA dehydrogenase (EC 1.3.99.3); long-chain acyl-CoA dehydrogenase (EC 1.3.99.13) and very long-chain acyl-CoA dehydrogenase. They all donate electrons to ETFQOR via ETF, and therefore contribute to the mitochondrial superoxide formation, which after conversion to H_2_O_2_ serves as a redox signaling. As with the β-like oxidation of BCKAs, FA β-oxidation leads to elevated rates of mitochondrial superoxide formation, hence to redox signaling. The mechanism should be identical to that described above. Even under conditions of low glucose, the ETF/ ETFQOR redox relay to complex I of the mitochondrial respiratory chain plus the ETFQOR system serves as the electron acceptor for dehydrogenases of β-oxidation. Interestingly, GPR40 signaling also activates NOX2 [367].

### 6.3. Redox-Sensitive Mitochondrial Phospholipase iPLA2γ Amplifies FASIS

#### 6.3.1. FASIS at Low Glucose vs. Dependence of Metabolic and Receptor Part of FASIS

There is the question of whether the GPR40-receptor-dependent component of FASIS also depends on the opening of Ca_V_ channels. This is equivalent to another question: whether the GPR40-receptor component of FASIS is absolutely dependent on the metabolic (mitochondrial) component. Since there are action potential spikes due to Ca_V_ opening upon GPR40 activation under low glucose conditions [340], we should conclude that there is at least experimental FASIS under low glucose conditions, and that the established metabolism under these conditions is sufficient to keep certain GPR40 downstream pathways operating and is thus able to stimulate insulin secretion. Speculatively, β-oxidation should provide conditions of elevated ATP plus H_2_O_2_ to trigger the “basic” portion of the IGV exocytosis, while the additional amplification provided by GPR40 should operate via the pathways described above in Section 6.2.1.

#### 6.3.2. Redox-Sensitive Mitochondrial Phospholipase iPLA2γ Amplifies FASIS

Notably, the 1st phase of experimental FASIS was recognized (and the second only partly) to be amplified by the action of mitochondrial phospholipase iPLA2γ, which provides the cleaved mitochondrial FAs to GPR40 in insulinoma INS-1E cells [309] and in mouse islets (Holendová et al., unpublished). iPLA2γ is directly activated by H_2_O_2_, hence it is also activated by the intramitochondrial redox signaling originating from FA β-oxidation. As a result, free long-chain FAs entering β-cells via CD36 and those cleaved from intracellular MAGs self-accelerate the GPR40 signaling and hence insulin secretion, after being imported into mitochondria and metabolized by β-oxidation. The activated iPLA2γ cleaves free long-chain FAs from mitochondrial phospholipids. The cleaved long-chain FAs diffuse to the plasma membrane and subsequently stimulate GPR40 (Figure 8). It should be investigated whether this is only allowed after their flip onto the outer phospholipid bilayer leaflet of the plasma membrane, i.e., onto their exterior. The plausibility of this event was suggested by an experiment in which activated iPLA2γ cleaved long-chain FAs, while their diffusion to the plasma membrane was indicated with the FA-sensitive fluorescent protein ADIFAB [309].

### 6.4. FASIS at High Glucose

#### 6.4.1. Distinction from FASIS at Low Glucose

It is not surprising that in the presence of both FA and higher glucose concentrations (those stimulating GSIS when applied without FAs), the fraction of metabolic triggering increases. A more intensive intermittent Ca_V_ opening with probably prolonged bursting may be a cause, which would induce an additional Ca^2+^ release from stores such as the ER, IGVs or lysosomes. At the same time, the GPR40-pathway stimulation of IGV exocytosis is likely to be maintained. These speculations are supported by so far the largest amount of insulin being secreted by INS-1E cells when palmitic acid plus 25 mM glucose were added together, relative to either stimulation by palmitic acid alone or GSIS alone [338]. These aspects require further experimentation in vivo to more closely resembled physiological conditions.

#### 6.4.2. β-Oxidation vs. Triglyceride/Fatty Acid Cycle in Pancreatic β-Cells

We discussed whether high glucose is essential for FASIS as such. Vice versa, a certain minimum level of FA metabolism was found to be required even for GSIS. This view was supported by experiments in which the inhibition of triglyceride lipolysis [368,369] attenuated GSIS in isolated PIs and the 1st GSIS phase in mice with deleted lipase specifically in β-cells [370] or in ATGL knockout mice [349]. However, the net FASIS induced by palmitate was not affected by the ATGL deletion synthesis of triglycerides being alternated with triglyceride hydrolysis [20]. Such cycling, termed the triglyceride/fatty acid cycle is futile, since it consumes ATP. Nevertheless, one may speculate that since DAG is one of the intermediates of the cycle, it may provide all the benefits described above in previous sections. Moreover, free FAs released during the cycle may add supplemental FASIS to the net GSIS.

Indeed, the addition of FAs to β-cells and PIs in the presence of insulin-stimulating glucose amplifies GSIS, while β-cell acyl-CoA levels increase and appear to rapidly esterify glycerol 3-phosphate into lysophosphatidic acid and several different glycerolipids [308]. Note that this partly replenishes cytosolic NAD^+^. Additionally, glycerol-3-phosphatase produces glycerol and thus regulates glycolysis, the cellular redox state, ATP production and other important branches of metabolism [371]. The consequences of higher glycerol-3-phosphate formation lie in the concomitant resulting increase in long-chain saturated MAGs [372,373]. These MAGs additionally stimulate insulin secretion via the GPR119 receptor and downstream PKA and EPAC2 pathways, the latter notably facilitates IGV priming by activating the protein Munc13-1 [372]. Note that this is similar to the GPR40 activation by the FAs cleaved in the cell interior, i.e., from the mitochondrial membranes by the mitochondrial phospholipase iPLA2γ.

In INS-1E cells, about two-thirds of the 1st phase of FASIS was dependent on GPR40 and nearly the same 33% of the maximum FASIS amplitude was observed upon silencing iPLA2γ (or its ablation in mice, unpublished). The remaining FASIS can be considered to be dependent on the elevation of ATP plus redox signaling from β-oxidation.

Higher glucose also decreases acyl-CoA levels in pancreatic β-cells [62]. Previously, acyl-CoAs were suggested to activate K_ATP_ [310], hence declining acyl-CoAs would ease K_ATP_ closure. As we described in detail above, the metabolism of the remaining long-chain acyl-CoAs leads to superoxide/H_2_O_2_ formation, which aids the opening of Ca_L_. In contrast, incoming higher glucose in pancreatic β-cells increases malonyl-CoA [62,374], which inhibits CPT1 and hence FA β-oxidation. This in turn opens the way for FA synthesis stemming from the ACL reaction after the citrate efflux from mitochondria [373]. Nevertheless, the silencing of ACL and fatty acid synthase in β-cells did not affect GSIS [311].

### 6.5. GLP-1 as an Important Stimulus for Postprandial Insulin Secretion after High Fat Meal

One can hypothesize that during fasting the increased FAs bound to albumin in circulation contribute to the ongoing basal secretion of insulin upon fasting [27]. However, it is necessary to investigate whether these minimum, non-pathological levels of FAs bound to plasma albumin possess the ability to stimulate a “basal” FASIS. Nevertheless, the ability to activate GPR40 by the FAs-bound to albumin was questioned [375,376]. Hypothetically, non-pathological levels of FAs in circulation upon fasting may stimulate GLP-1 secretion in intestinal enterocyte L-cells. The resulting basal GLP-1 can therefore stimulate a certain level of insulin secretion in pancreatic β-cells. This speculation will be plausible once there is definitive evidence that GLP-1 stimulates insulin secretion under low glucose conditions.

In contrast, upon high glucose intake, until chylomicrons arrive, albumin-bound FA levels are minimum, because the lipolysis is simultaneously inhibited in white adipose tissue due to the blockage of adipocyte lipases upon ongoing insulin receptor signaling, and so glucose uptake to adipocytes predominates [377]. The timing of the occurrence of GLP-1 in human circulation exhibits a peak of <10 pmol/L [267] between 30 and 60 min after carbohydrate or protein intake, and 120 min after the ingestion of lipids [274]. When compared to the availability of chylomicrons (see above), responses to GLP-1 should precede those of chylomicrons. However, their overlap is not excluded.

### 6.6. Other Sectretagogues

The arginine-mediated amplification of insulin secretion was previously reported to be independent of nitric oxide synthase [378]. Arginine effects related to insulin secretion are also mediated by the metabotropic receptor GPR6A [379]. Additionally, adenylosuccinate (S-AMP) was found to rise in response to glucose, and may be considered to be a secretagogue [380]. The Conversion of IMP to S-AMP is provided by the adenylosuccinate synthase. S-AMP then activates the sentrin/SUMO-specific protease SENP1 [381], which influences redox homeostasis similarly to the glutathione/glutaredoxin system [13].

## 7. Mechanisms of Transfer of Redox Signaling and Possible Insults

### 7.1. Mechanims of Transfer of Redox Signaling

Mechanisms of redox signal conduction within the cell can either rely on a simple H_2_O_2_ diffusion or on the redox relay mediated by thiol-based proteins that reversibly form S-S bridges and are capable of transferring such a transient oxidation to the target [14]. Peroxiredoxins are such proteins, and their oxidation states are regenerated via thioredoxins and glutaredoxins. Redox signal transfer over a distance, which can be as far as several 100 nm, would significantly decay when conducted by simple H_2_O_2_ diffusion, due to non-specific reactions with random targets within the cytosolic (membrane) milieu. In contrast, the peroxiredoxin oxidized into the first stage can diffuse and transfer the redox signal by oxidation of the distal protein target [14]. It could be speculated that a mutually interacting array of peroxiredoxin conducts the redox signal instead of their diffusion.

Additionally, the so-called floodgate model could explain the spreading of a redox signal over a larger distance [1,15]. The model is based upon the ability of peroxiredoxin to be oxidized into sulfenyls, sulfinil and sulfoxyl stages. Moreover, these two highly oxidized stages form clusters of regular decamers. A local H_2_O_2_ source then oxidizes peroxiredoxin decamers, which do not escape from its proximity, into their high oxidized states. As a result, the only ones that escape by diffusion are kept in the lowest oxidized stages and can oxidize the proper specific target in a sufficient distance. Experimentally, targets such as a signal transducer and activator of transcription 3 (STAT3) or apoptosis signal-regulating kinase 1 (ASK1) were confirmed to react with peroxiredoxin 2 and 1, respectively, in this way [382,383,384,385]. Targets relevant for GSIS and their interactions with peroxiredoxin and/or H_2_O_2_ are yet to be determined, as discussed in Section 2.3.1 and Section 2.3.2.

Note, however, that for high levels of H_2_O_2_, its detoxification is ensured by the peroxiredoxin/thioredoxin antioxidant system [386,387]. As was already mentioned in the Introduction, the glutathione content is relatively low in pancreatic β-cells [8,9,10,11]. Since in contrast, the relative content of thioredoxins and glutaredoxins [10,11], and peroxiredoxins and other proteins capable of redox relay is satisfactorily high, the above-mentioned redox-relay spread of redox signals is easily conducted in such a low-glutathione milieu [14,15]. In conclusion, the pancreatic β-cell employs a plethora of redox signals for the acute stimulation of insulin secretion. The appearance of those signals (or their frequency) may hypothetically maintain the correct transcriptome to maintain β-cell identity [4,17].

### 7.2. Redox Status of Pancreatic β-Cells may Affect and Impact Redox Signaling Essential for Insulin Secretion

Dor and colleagues hypothesized that repeating K_ATP_-dependent insulin exocytosis and all ionic and regulatory phenomena involved also contribute to the maintenance of the insulin gene expression, hence to the maintenance of so-called β-cell identity [17]. Subsequently, redox signaling, which is essential for GSIS and also for the stimulation of insulin secretion by other secretagogues (BCKAs and FAs), is also expected to be one of the key factors to maintain β-cell identity [16]. These aspects require further research. Incorrect checking of β-cell identity (meaning dysregulation of proper signaling maintaining proper expression of β-cell-specific genes and suppression of so-called “disallowed” genes) may subsequently lead to particular stress signals to the periphery, such as those initiating development of peripheral insulin resistance. This progresses to type-2 diabetes development. Indeed, we found that a single gene ablation, ablation of NOX4 responsible for the majority of redox signaling upon GSIS, generates the onset of insulin resistance [4]. We could speculate that in this case, pancreatic β-cells must emit an as of yet unidentified stress signal (directly or via the immune system to the periphery) just because of the insufficient identity checking or autocrine self-maintenance of β-cells in mice with ablated NOX4.

It has long been assumed that pancreatic β-cells are particularly vulnerable to oxidative stress because of the relatively low expression of several antioxidant enzymes, including catalase, superoxide dismutase and GSH peroxidase [8,388]. However, recent findings identify the peroxiredoxin/thioredoxin antioxidant system as the major detoxifying pathway of β-cells [386,387]. Thus, it has been shown that β-cells have the capacity to detoxify micromolar levels of H_2_O_2_ through an endogenous thioredoxin reductase–dependent mechanism, a finding that can be masked by the use of nonphysiological bolus delivery of oxidants to the experimental system [386]. Given the catalytic efficiencies of peroxiredoxins for H_2_O_2_ (10^5^ M^−1^ s^−1^) [389], the affinity of peroxiredoxins for H_2_O_2_ (K_m_ < 20 μM) and the property of peroxiredoxins to be deactivated through oxidation, it has been suggested that peroxiredoxins are the prime candidates for mediators of H_2_O_2_ signaling in addition to detoxification in β-cells [14,386,387]. Based on these suggestions, we hypothesized that the NOX4-produced and mitochondria-generated H_2_O_2_ at GSIS in vivo could also be maintained in balance by the peroxiredoxin/thioredoxin antioxidant system. Oxidative stress that contributes to diabetic etiology occurs when such redox homeostasis is impaired.

Additionally excessive antioxidant protection may harm basic processes in pancreatic β-cells [4]. For example, cytosol-targeted antioxidant therapy would inevitably suppress GSIS due to (partial) elimination of redox signaling. Such therapy should be inappropriate even for the early stages of diabetes. Tuning down the essential H_2_O_2_ release at GSIS would instead amplify prediabetes symptoms instead of preventing them. In contrast, we predicted that mitochondria-targeted antioxidants should not harm the physiological redox signaling (except that of BCKAs and FAs) and may be better tolerated when the premature oxidative stress in the matrix at the prediabetic stage is treated in this way [4].

The term mitochondrial hormesis, or mitohormesis, has been introduced to describe a defense mechanism observed in oxidant-induced stress-responses by mitochondria [390]. Persistent decrease of mitochondrial oxidative phosphorylation and the activity of respiratory complexes are associated with the release of oxidants from nonmitochondrial sources, together with the release of proinflammatory and profibrotic cytokines, and a manifestation of organ dysfunction. Consequently, restoration of mitochondrial function and superoxide production via activation of AMPK has been associated with improvement in markers of renal, cardiovascular and neuronal dysfunction with diabetes [391].

In addition, putative redox-active compounds, such as dietary antioxidants and olive oil nutraceuticals, are being studied as potential regulators of β-cell redox homeostasis, insulin secretion, blood glucose homeostasis and the development of type 2 diabetes (for selected recent reviews, see [392,393,394,395,396]). Considering the mechanism of action of dietary phytochemicals, we would like to alleviate the general belief that these compounds act as direct free radical scavengers in vivo. Kinetic constraints, given by the peroxiredoxin/thioredoxin antioxidant system in β-cells, indicate that in vivo one-electron scavenging of radicals by dietary antioxidants is ineffective in antioxidant defense. Enzymatic removal of non-radical electrophiles, such as hydroperoxides, in two-electron redox reactions is the primary antioxidant mechanism [397]. It has been proposed that a major mechanism of action for nutritional antioxidants is the paradoxical oxidative activation of the NRF2 (NF-E2-related factor 2) signaling pathway, which maintains protective oxidoreductases and their nucleophilic substrates [397,398,399]. The nutritional compounds are thus redox-active by activating signaling pathways, such as NRF2, and mimic the effect of endogenously produced electrophiles, a mechanism termed parahormesis [398]. Nevertheless, the detailed description of the role of nutrition, dietary polyphenols and their relation to the etiology of type 2 diabetes is beyond the scope of this review. A reader can refer to our previous review and to other relevant reviews published on the topic [16,396,400,401,402].

## Figures and Tables

**Figure 1 antioxidants-10-00197-f001:**
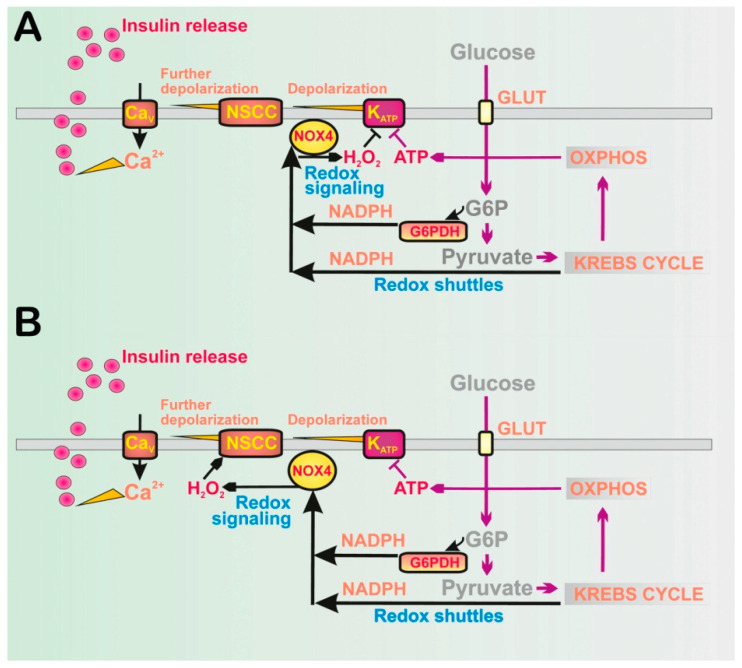
A new paradigm in the mechanism of glucose-stimulated insulin secretion (GSIS): Essential requirement of redox signaling. Upon glucose intake, pentose phosphate pathway (PPP) and redox shuttles supply cytosolic NADPH to increase NOX4 activity and thus elevate H_2_O_2_, which substantiates redox signaling. The two possible hypotheses assume two distinct targets of H_2_O_2_: either that (**A**) K_ATP_ is closed exclusively when both ATP plus H_2_O_2_ are elevated (as supported by the patch-clamp data in Ref. [4]); or (**B**) that K_ATP_ is closed by ATP, whereas the TRPM2 channel (or other nonspecific calcium channels (NSCCs)) is activated by H_2_O_2_. In both cases, the depolarization shift by NSCCs is essentially required for achieving the −50 mV depolarization threshold for Ca_V_ activation despite 100% of the K_ATP_ ensemble being closed. Mechanism (**A**) plus additional H_2_O_2_ stimulation of TRPM2 is also plausible.

**Figure 2 antioxidants-10-00197-f002:**
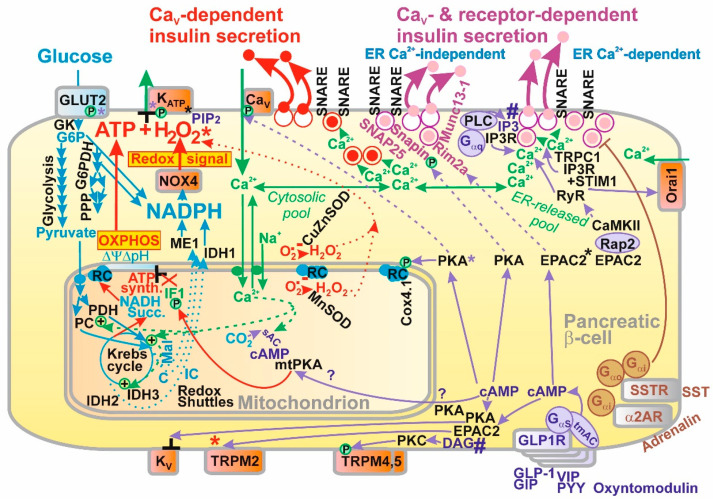
A new paradigm in the mechanism of GSIS in the context of the regulation of insulin secretion. Glucose transporters, GLUT2 in rodents and GLUT1 in humans, allow the equilibration of blood glucose with the cytosolic glucose concentration in pancreatic β-cells at any time. A specific hexokinase isoform, glucokinase (GK) is not feedback inhibited, which together with the absence of functional pyruvate dehydrogenase (PDH) kinases and lactate dehydrogenase, allows the entire glycolytic output to be utilized for oxidative phosphorylation (OXPHOS) and hence efficient synthesis of ATP. As a result, elevated blood glucose induces elevated ATP synthesis and respiratory chain (RC) proton pumping, thus increasing ΔΨ_m_ and respiration. Moreover, a portion of glucose-6-phosphate formed by GK enters the PPP, producing NADPH at two sites, by G6P dehydrogenase (G6PDH) and 6-phosphogluconate dehydrogenase. This elevates cytosolic NADPH and thus the NOX4 activity. Since NOX4 directly produces H_2_O_2_, elevated glucose evokes instant redox signaling. Hypothetically, either K_ATP_ is only closed when both ATP and H_2_O_2_ are elevated and TRPM2 can also be independently activated by H_2_O_2_ (red *); or ATP only closes K_ATP_ and H_2_O_2_ activates TRPM2, as one of NSCCs, which is essentially required to reach the threshold of −50 mV for the activation of Ca_V_. The intermittent opening of Ca_V_ channels followed by the opening of K_V_ leads to repeatable action potential spikes, provoking pulsatile Ca^2+^ entry and oscillations in the cytosolic Ca^2+^ concentrations, which stimulate the exocytosis of insulin granules. There are amplifying regulation systems: (i) given by the Ca^2+^ import to the mitochondrial matrix, where increased Ca^2+^ activates several dehydrogenases, including the PDH complex; and activates the PKA-mediated phosphorylation of Cox4.1 and sAC-mtPKA-mediated phosphorylation of IF1, where the former protects the ATP inhibition of Cox4.1, and the latter releases the inhibition of the ATP-synthase and (ii) redox shuttles contribute to the elevated cytosolic NADPH. Incretins, GLP-1 or GIP, massively amplify insulin secretion via their receptors, transferring signals via the Gαs proteins, tmAC, cAMP and either the PKA or EPAC2A pathway. PKA increases the sensing range of K_ATP_ to physiological mM ATP, inhibits K_V_, activates Ca_V_, GLUT2 (gray *) and snapin. EPAC2A directly activates TRPM2, essential for KATP-triggered GSIS; and activates Rim2α or via Rap2-calmodulin kinase II (CaMKII) activates RyR, providing the Ca2+efflux from ER, when CaL is open, thus again amplifying IGV exocytosis. Biased GLP1R (GIPR) signaling activates PLC and either IP3-IP3R or PLC STIM1-Ora1-TRPC1 mediated Ca2+ efflux from the ER (or TRPC1 migration to the plasma membrane); or the DAG-PKC pathway phosphorylating and thus activating TRPM4 and 5, so providing another essential NSCC-mediated shift up to −50 mV. Somatostatin (SST) and the andrenergic inhibitory pathway (via Gi or Go proteins) are also depicted.

**Figure 3 antioxidants-10-00197-f003:**
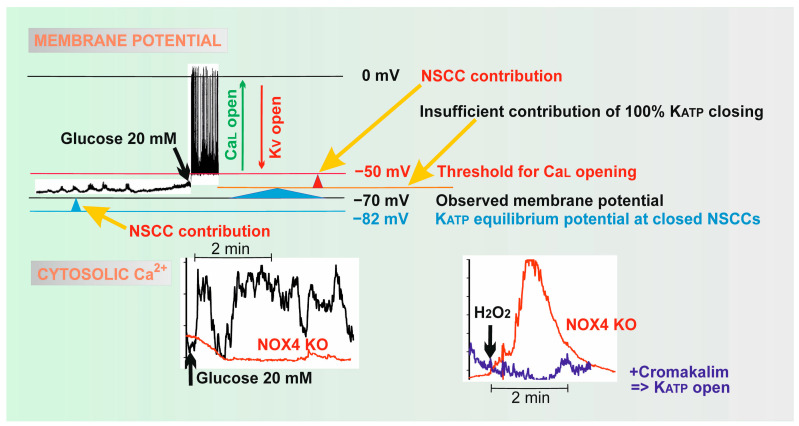
Schematics of parallel plasma membrane potential action potential vs. Ca^2+^ oscillations in pancreatic β-cells after the addition of 20 mM glucose. Data similar to that published in Ref. [4] are used to illustrate: plasma membrane action potential firing (top) and Ca^2+^ oscillations indicated with FURA2 in INS-1E cells silenced for NOX4 (red trace “NOX4 KO”) as compared with the Ca^2+^ oscillations rescued in silenced cells by H_2_O_2_ addition (bottom). The latter were prevented by cromakalim, keeping K_ATP_ open. The observed resting plasma membrane potential accounts for −70 mV in contrast to the theoretical potential of −82 mV at equilibrium (blue line), if K_ATP_ was contributing exclusively. The shift is caused at least by the non-specific Ca^2+^ channels (NSCCs), the fluxes of which enable depolarization up to −50 mV (red line), above which the activation of Ca_V_ channels proceeds. The intermittent Ca_V_ channel (and Na^+^ channel) opening followed by the K_V_ action creates the action potential spikes. Without the contribution of NSCCs, the 100% closed ensemble of K_ATP_ channels would only reach a depolarization (orange line) that is insufficient for the activation of Ca_V_. Since Ca^2+^ oscillations cease when H_2_O_2_ was not supplied by NOX4 (red trace), one can explain the essential requirement of redox signaling by the two possible above-mentioned hypotheses that differ in the two distinct targets of H_2_O_2_: (**A**) either that K_ATP_ is closed exclusively when both ATP plus H_2_O_2_ are elevated (and TRPM2 can be independently also activated by H_2_O_2_); or (**B**) that K_ATP_ is closed by ATP, whereas the TRPM2 channel (or other NSCCs) is activated by H_2_O_2_. In both cases, the depolarization shift by NSCCs is essentially required for achieving the −50 mV threshold for Ca_V_ activation, despite 100% of the K_ATP_ ensemble being closed. Patch-clamp results tend to support model (**A**) [4].

**Figure 4 antioxidants-10-00197-f004:**
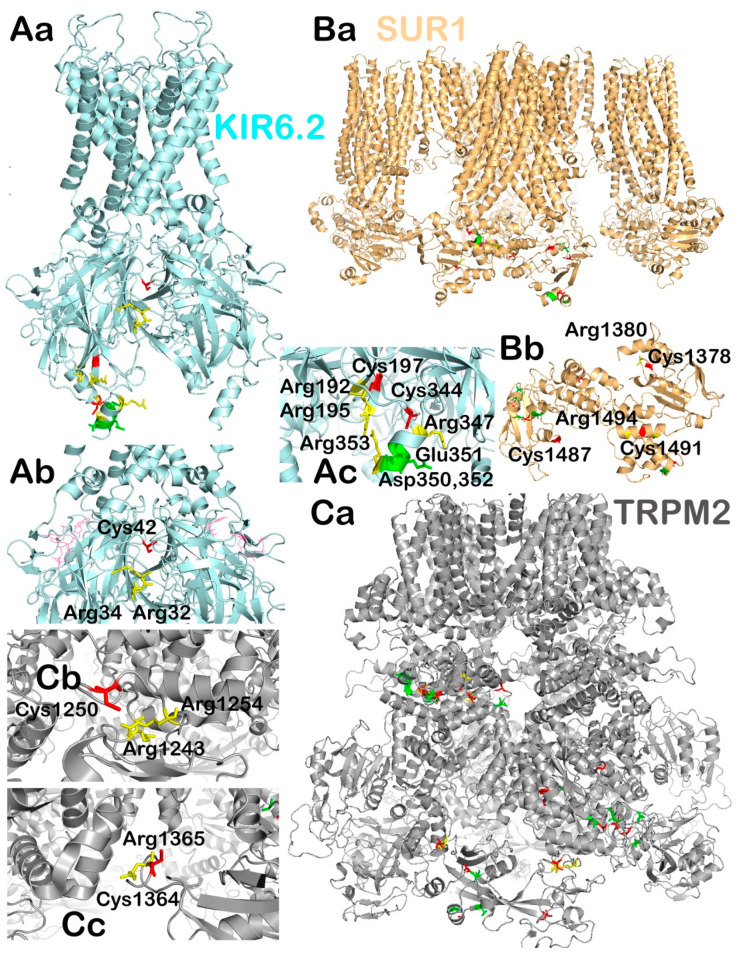
Predicted cysteines interacting with H_2_O_2_ in KIR6.2, SUR1 and TRPM2 structures in pancreatic β-cells. Synthetic structure of mouse/hamster pancreatic ATP-sensitive potassium channel (pdb code 5wua) [133] and human TRPM2 bound to ADPR and calcium (pdb code 6pus) [157] were used for the visualization of the potential redox reactive cysteines. (**Aa**,**Ab**,**Ac**) Tetrameric assembly of KIR6.2 subunits (light blue) and details of potentially redox regulated Cys with surrounding Arg and/or Asp and Glu, modulating the Cys reactivity towards redox modifications in intracellular regions of the channel; (**Ba**,**Bb**) prediction of hypothetical redox regulated Cys within one intracellular domain of the SUR1 subunit (light orange) with Arg and Asp/Glu in close proximity and (**Ca**,**Cb**,**Cc**) human TRPM2 (light grey) and detailed visualization of hypothetical redox regulated Cys in intracellular domains; color coding: Cys in red, Arg in yellow, Asp and Glu in green.

**Figure 5 antioxidants-10-00197-f005:**
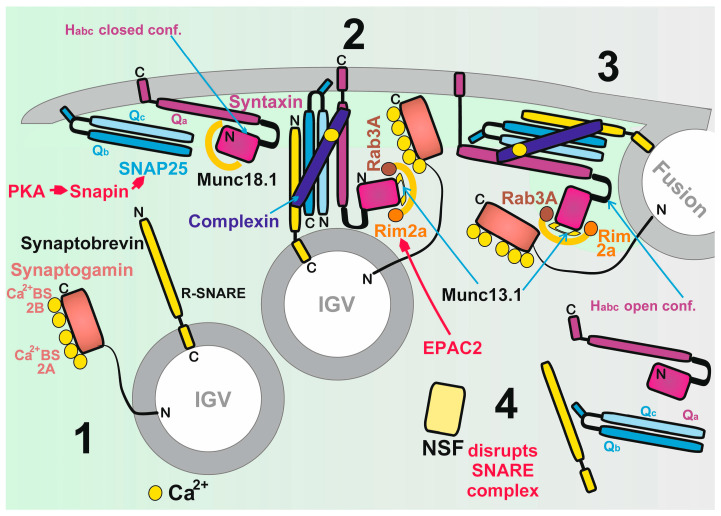
Mechanisms of insulin granule vesicle (IGV) exocytosis, schematically illustrated in four stages: Ca^2+^ ions are depicted as yellow circles. C- and N-ends of proteins are marked and specific protein domains. Stage 1, when the Munc18.1 protein binds the Habc domain of syntaxin-1, locking it in the closed conformation. The elevated Ca^2+^ promotes Ca^2+^ binding to the synaptogamin C2 domains at the Ca^2+^ binding sites 2B and 2A (“Ca^2+^ BS 2B and 2A”) and to the C2 domain of the complexin. In stage 2, priming by the Rim2α and Rab3A GTPases and their interaction with the Munc13.1 alters the syntaxin-1 conformation to the open one. This allows the formation of a large SNARE complex, the four-component bundle coiled-coil SNARE complex. In stage 3, SNARE complexes in a zippering fashion have enough energy to move the vesicle into close proximity to the plasma membrane and ensure fusion. At first, a fusion stalk (not shown) is formed followed by a larger septum until the content of the IGV lumen relocates to the outside of the cell. Along the pore expansion, the *cis* to *trans* conformation change of the SNARE complex takes place. In the terminating stage 4, the large SNARE complex is disrupted by the NSF ATPase. The verified PKA and EPAC2 targets are also shown. Note, that certain synaptogamin isoforms such as isoforms 4 and 8 do not bind Ca^2+^. If these were present in β-cells, this would allow IGV exocytosis at lower Ca^2+^ concentrations. However, Ca^2+^ would still be required for complexin.

**Figure 6 antioxidants-10-00197-f006:**
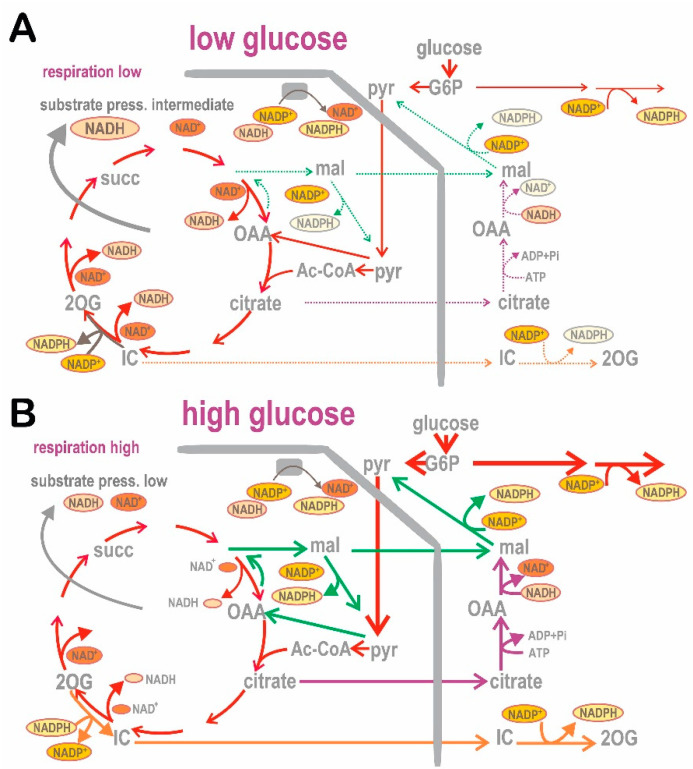
Redox shuttles—(**A**) operate at minimum in low glucose conditions, but (**B**) at maximum in the high glucose conditions. The key enzyme for the pyruvate/malate shuttle (green arrows) is the pyruvate carboxylase (PC) allowing the reverse malate dehydrogenase (MDH) reaction and following the export of malate (Mal) to the cytosol, where malic enzyme 1 (ME1) forms NADPH. As a result, instead of NADH, which would otherwise be made by MDH, a cytosolic NAPDH is formed. Similar consequences result from the operation of the pyruvate/citrate shuttle and pyruvate/isocitrate shuttle (“IC”, isocitrate). The latter involves NADPH-dependent reductive carboxylation by isocitrate dehydrogenase 2 (IDH2) in the mitochondrial matrix, followed by the export of the formed IC, and oxidative decarboxylation by the cytosolic IDH1, forming NADPH, which again contributes to the elevated cytosolic NADPH pool. Note that the operation of these shuttles under conditions of high glucose excludes the operation of the malate/aspartate, which requires exactly the opposite 2-oxoglutarate fluxes.

**Figure 7 antioxidants-10-00197-f007:**
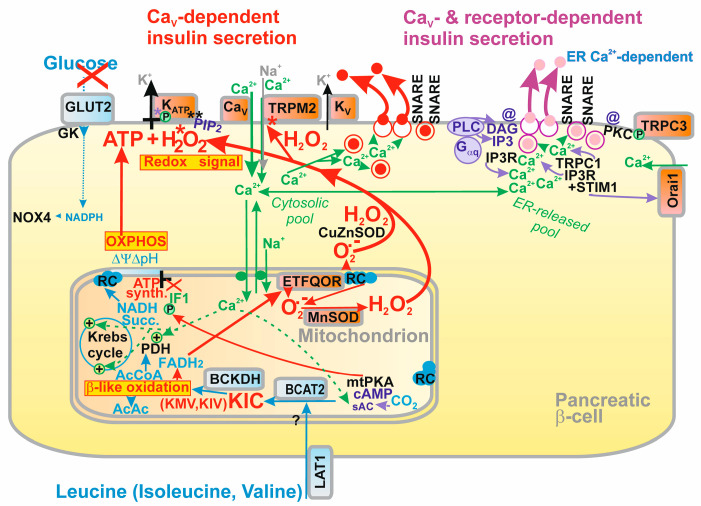
The mechanism of branched-chain keto acid-stimulated insulin secretion involves redox signaling of mitochondrial origin. 2-ketoisocaproate (KIC), 2-ketomethylvalerate (KMV) and 2-ketoisovalerate (KIV) resulting from leucine, isoleucine and valine (imported into cells by the plasma membrane LAT1 transporter), respectively, due to the branched-chain aminotransferase reaction in the mitochondrial matrix (BCAT2), are metabolized by branched-chain ketoacid dehydrogenase (BCKDH), both existing exclusively in the mitochondria of β-cells. A series of reaction of β-like oxidation begins at the dehydrogenase step, yielding FADH_2_. This dehydrogenase cofactor is reoxidized by the ETF, as a single electron carrier, from which ETFQOR accepts electrons and converts Q to QH_2_. As a side reaction, superoxide is formed by the respiratory chain or even within the ETFQR. In the mitochondrial matrix, superoxide is transformed to H_2_O_2_ by MnSOD, whereas by CuZnSOD in the intermembrane space and cytosol. The elevated mitochondrial/cytosolic H_2_O_2_ substitutes the redox signaling of the NOX4 origin. Consequently, such redox signaling, together with elevated ATP, sets the triggering conditions for insulin secretion. Either both H_2_O_2_ and ATP evoke the K_ATP_ closure (and TRPM2 is independently activated by H_2_O_2_), or K_ATP_ closing is “classically” ensured by the elevated ATP, while H_2_O_2_ activates TRPM2 or other NSCCs provide the essential NSCC-mediated shift to the threshold depolarization. As a result, Ca_V_ channels are open, providing the Ca^2+^ signal for IGV exocytosis. The end products of the β-like oxidation of KIC, contributing to OXPHOS, are acetoacetate (AcAc) and acetyl-CoA (AcCoA). AcCoA enters the Krebs cycle to form citrate. AcAc can be partly exported to the cytosol or is converted to β-hydroxybutyrate by a reverse reaction of β-hydroxybutyrate dehydrogenase at the expense of NADH. Moreover, since theoretically the amplifying mechanism should remain intact, we predict that also the net (experimental) ketoacid-stimulated insulin secretion should be amplified by GLP1 and GIP (see Figure 2).

**Figure 8 antioxidants-10-00197-f008:**
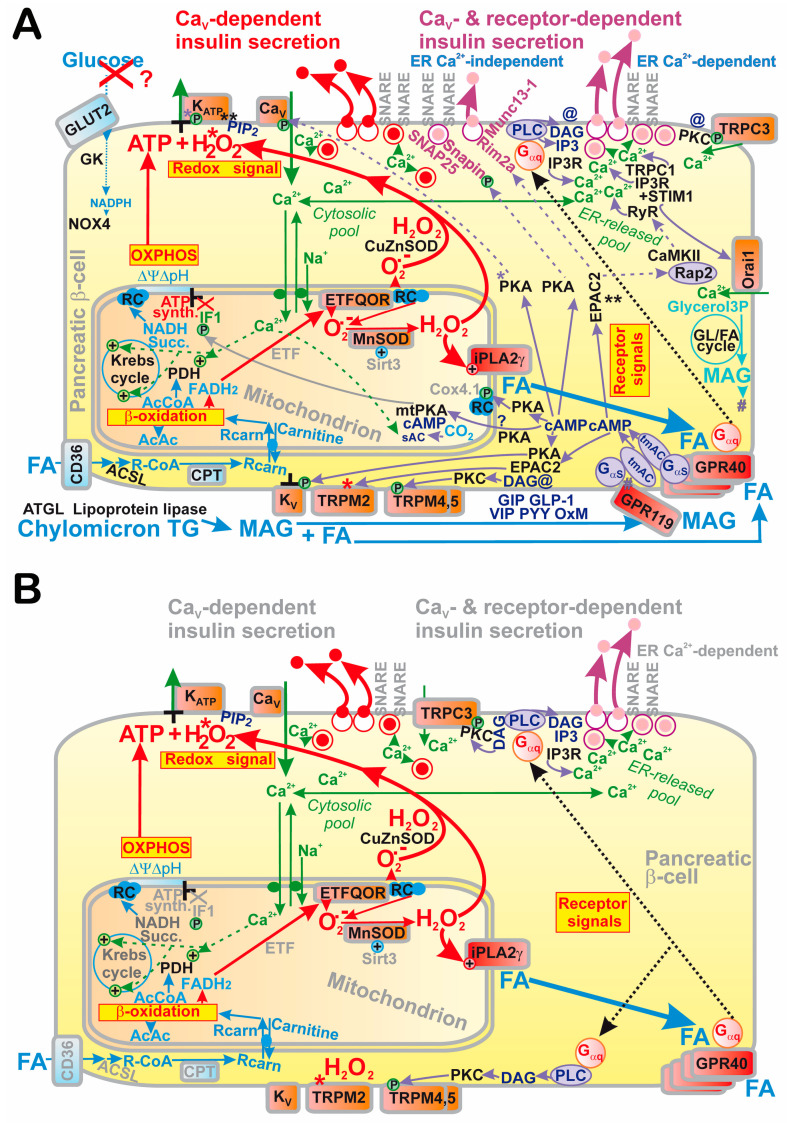
(**A**) Complex mechanism of FASIS and (**B**) GPR40 receptor part. FASIS is predominantly mediated by the GPR40 receptor. There are controversies over whether FAs bound to serum albumin and those cleaved by lipoprotein lipase can be transported into β-cells by FA transporter CD36 under fasting conditions. Such FA transport and cleavage of triglycerides (TG) to monoacylglycerols (MAG) and FA, however, take place when postprandial chylomicrons reach the rich capillaries of the pancreatic islets. MAG subsequently activates the metabotropic receptor GPR119, while long-chain FAs activate GPR40. The activations of both receptors lead to Ca_V_-mediated action potential spikes and concomitant pulsatile insulin secretion. The canonical GPR119 signaling and biased GPR40 signaling leads to cAMP activation of the PKA and EPAC2 pathways with all the consequences described in the legends to Figure 2 and Figure 7. Our in vitro and in vivo experiments with mice (unpublished) demonstrated that approximately 2/3 of the GPR40 response is given by the amplifying mechanism due to the mitochondrial phospholipase iPLA2γ/PNPLA8, similarly as in INS-1E cells [309]. The phospholipase iPLA2γ is directly activated by the elevated H_2_O_2_ in the mitochondrial matrix and cleaves both saturated and unsaturated FAs from the phospholipids of mitochondrial membranes. The cleaved free FAs diffuse up to the plasma membrane, where they activate GPR40. FASIS in iPLA2γ knockout mice or its isolated islets yields 30% insulin in the 1st fast phase of insulin secretion, compared to wt mice (Holendová et al., unpublished). This supports the existence of such an acute mechanism in vivo. Besides the GPR40 pathway, a portion of FASIS still results from FA β-oxidation, with a similar mechanism as was described for ketoacids (see the Figure 7 legend). The difference lies in the several reaction turns during β-oxidation and a higher yield of FADH_2_/QH_2_ and AcCoA (and, of course ATP) from a single FA molecule. Theoretically, the stimulation of incretin receptors and PKA (and other kinase) pathways should also contribute to an amplification of responses to FAs. Nevertheless, the fraction of this contribution should be small due to a nearly overwhelming contribution of GPR40 and GPR119. Moreover, when FAs stimulate insulin secretion together with high glucose, the contribution of the so-called triglyceride/FA cycle, alternatively termed the glycerol-3-phosphate fatty acid cycle (G3P/FA cycle), also amplifies insulin secretion. This is provided by creating more ligands for the GPR40 and GPR119 receptor and by additional DAG signaling.

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
