# Peer review of "The Pancreatic β-Cell: The Perfect Redox System"

_antioxidants, 2021, doi:10.3390/antiox10020197_

Round 1

Reviewer 1 Report

The review written by Jezek and al. focuses on redox system in beta-pancreatic cells. this review represents a considerable amount of work. It is well illustrated, documented and truly unique. Of course it must be published. It will be a reference in the field. However, there are a few minor points to be improved.

Minor Points

Figures 1 and 5: changing the yellow writing and arrows to another color

Figures 2, 7 and 8: a small simplification would make them lighter and more understandable. The elements related to the illustrated paragraph should be clearly visible by removing the other elements or making them less visible. It is sometimes difficult to find the points that these figures are supposed to illustrate.

Author Response

We thank the reviewer for appreciation of our review. In the revised version, we did change yellow writing and arrows to another color (mostly black). We did simplify revised Figure 7 and we present figure in the full and simplified form as Figs.8A and B. Thus Figure 8B shows exclusively GPR40 receptor signaling, whereas Fig8A shows all what happens during incoming triglycerides into beta cell capillaries.   

We cannot simplify Fig.2 , since it had already been simplified as Fig.1A or B...It woudl have no sense to show the same figure twice.

Reviewer 2 Report

This is a well written and cohomprensive review, which will raise the interest of those working in the field. In my opinion, it could be accepted for publication in Antioxidants in its present form

Author Response

We thank the reviewer for appreciation of our review.

Reviewer 3 Report

Lifestyle is the primary prevention of diabetes, especially type-2 diabetes (T2D). Nutritional intake of olive oil (OO), the key Mediterranean diet component has been associated with the prevention and management of many chronic diseases including T2D. Several OO bioactive compounds such as monounsaturated fatty acids, and key biophenols including hydroxytyrosol and oleuropein, have been associated with preventing inflammation and cytokine-induced oxidative damage, glucose lowering, reducing carbohydrate absorption, and increasing insulin sensitivity and related gene expression. In addition, oleuropein (Ole), the main bioactive phenolic component of Olea europaea L. has recently attracted the scientific attention for its several beneficial properties, including its anticancer effects,  decreasing melanoma cell proliferation and motility. OLEO was also able to reduce the rate of glycolysis of human melanoma cells without affecting oxidative phosphorylation. This reduction was associated with a significant decrease of glucose transporter-1, protein kinase isoform M2 and monocarboxylate transporter-4 expression, possible drivers of such glycolysis inhibition. Thus, polyphenols are strong antioxidants with characteristics that are of beneficial therapeutic values for their development as candidates targeting oxidant-induced diseases. Consistent with this notion, interplay and coordination of redox interactions with polyphenols is an emerging area of research interest in antidegenerative therapeutics. Moreover, hormetic responses occur in a wide range of biological models for a large and diverse array of endpoints. Particular attention has been given to providing an assessment of the quantitative features of the dose-response relationships and underlying mechanisms that could account for the biphasic nature of the hormetic response after exposure to redox active agents. The hormetic dose response should be seen as a reliable feature of the dose response for hormones, redox regulated transcriptional factors  as well as  antioxidant compounds and appears to have an important impact  on brain pathophysiology and stress resistance mechanisms to oxidative and inflammatory insult and neurodegenerative damage.

This is an interesting paper.  The study is well-conceived and well-executed. This reviewer is satisfied with the significance of this study, the care in which the study was performed, and the implications of the results for human health.  However, although the results presented are convincing, the work raises some concerns which will need to be addressed. The questions posed are of extremely high interest, but the paper does not give adequate definitive information, therefore pending addressing some major question is possible to accept for publication.

Major concerns:

1. The role of redox active compounds, such as nutritional antioxidants in altered Glucose metabolism and regulation of hyperglycemic insult should be considered and discussed appropriately.

2. Preconditioning signal leading to cellular protection through Hormesis is an important redox dependent mechanism operating in cell survival. This aspect should be highlighted in the discussion and reference properly added (Siracusa et al., Antioxidants 2020, 9(9):E824)

3. Given the relationship between polyphenol compounds, redox status and the vitagene network and its possible biological relevance in cytoprotection, Authors while interpetrating results should discuss appropriately this aspect and make proper connection with emerging principles of hormesis and Branched chain AA metabolism in health and, specifically, in diabetic glucose dysregulated disease.

Author Response

We now added a new section 7.2 while the former section 7 is now 7.1 and briefly amended on the topic requested by the reviewer.